# Peculiarities of Long-Term Changes in Air Temperatures Near the Ground Surface in the Central Baltic Coastal Area

**Agu Eensaar**

Centre for Sciences, Tallinn University of Applied Sciences, Pärnu mnt 62, 10135 Tallinn, Estonia; agu.eensaar@gmail.com

**Abstract:** The peculiarities of the long-term change of the annual and monthly average air temperatures until 2017 in five cities of the coastal area of the Central Baltic region—Stockholm, Tallinn, Riga, Helsinki, and Saint Petersburg—were studied. The anomalies of the annual and monthly average air temperatures in relation to the average characteristics 1961–1990 were analyzed. The trends in the air temperature changes during 1980–2017, which come to 0.5 °C per ten years, have been found in the cities of the Central Baltic coastal area. The average air temperature in the Central Baltic cities has grown faster than the global and northern hemisphere. For the longer period of 1850–2017, the average annual rise of air temperature was within the range of 0.1 °C per ten years. The rise in temperature in different months is different, and the rise of the of the average temperature in the summer period has not occurred (at a significance level of 0.05). With the analysis of the frequency distributions of the average annual air temperatures and Welch's t-test, it is demonstrated that the air temperature (at a significance level of 0.05) has risen in all the months only in Saint Petersburg during 1901–2017 in comparison to the 19th century. There has been no reliable rise of the air temperature during the century in February and from June to September in Riga, from June to October in Helsinki, from June to September in Stockholm, and in August and September in Tallinn. It was found that the average air temperature trends have a certain annual course. The air temperature has risen most in March and April, reaching 0.09 °C (Stockholm, Tallinn) up to 0.23 °C (Saint Petersburg) per ten years. From June to September, the rise of air temperature is considerably lower, remaining below 0.04 °C per ten years. The changes in air temperature are small during the summer and mid-winter; the air temperature has significantly risen in autumn and spring.

**Keywords:** air temperature; climate change; Welch's *t*-test; trends; Baltic area

## 1. Introduction

Climate is a variable component of the natural environment. It has always been changing in the past, it is changing nowadays and it will certainly change in the future. If different time periods are used for averaging, the mean values of climatic variables are usually not constant. Advances in the observations and data analysis of climate change can provide a clearer understanding of the inherent variability of the climate system and its response to human and natural influences [1].

Climate change has a considerable impact on the natural environment and human activities in general and it can differ by regions. Because of the relevance of this issue, the Intergovernmental Panel on Climate Change (IPCC) was established in 1988 by the World Meteorological Organization (WMO) and the United Nations Environment Programme (UNEP) with the aim to provide a comprehensive assessment of all aspects of climate change [2].

To forecast climate change and its impacts, the peculiarities in the changes of various components related to the climate need to be examined. One of the significant parameters characterizing climate

change is the air temperature. Global warming, as per recent studies [3–10], is one of the major global concerns. Therefore, it is essential to study the changes in air temperature at the local or regional level. Several studies dedicated towards the examination of climate change various regions have shown that the rise of air temperature is not a simple and linear process, but a rather complicated phenomenon [11–13]. Several studies have also been devoted to the study of climate change in the Baltic region, the results of which allow the nature and extent of changes to be assessed [14–30]. For instance, the results of the studies have shown that the average annual temperature in Estonia is growing much faster than globally. At the same time, the temperature rise is different at different months. The air temperature has risen in Estonia in winter and early spring; however, in the summer, no significant temperature rise has taken place [15].

An increase in air temperature is an important factor that causes a variety of changes of natural environment, such as snow cover duration and sea ice conditions, as well as changes in land and marine ecosystems [31,32]. Considering the above, it is extremely important to study temporal and spatial temperature changes in in the Baltic Sea region.

Meteorological observations have been performed for a long time in the countries surrounding the central part of the Baltic Sea. The analysis of the climatic data, in different measuring points, enables the assessment of the general trends and also the differences, which together have an impact on the Baltic Sea and the development of its ecosystem.

Besides the randomness, there are certain spatial and temporal peculiarities. The present study aims to analyze the regularities of the long-term changes in the average air temperatures near the ground in the Central Baltic coastal area.

## 2. Materials and Methods

The present study aims at investigating the peculiarities of the long-term changes in the average annual and monthly air temperatures of the five cities in the Central Baltic coastal area until 2017. The observed cities are Stockholm (Sweden), Tallinn (Estonia), Riga (Latvia), Helsinki (Finland), and Saint Petersburg (Russia) (Figure 1). Meteorological observations have been performed in these cities for a long time. Specific measuring points have changed in some cases, for example, they have moved further from the city center to eliminate the direct heat effect of the city. In case of performance of the parallel observations, and in consideration of other circumstances, most of the data have been reduced to a specific location (harmonized). Measuring instruments and methods have also developed and changed over time, and therefore, the accuracy of the earlier data is lower than that of the measurements performed nowadays.

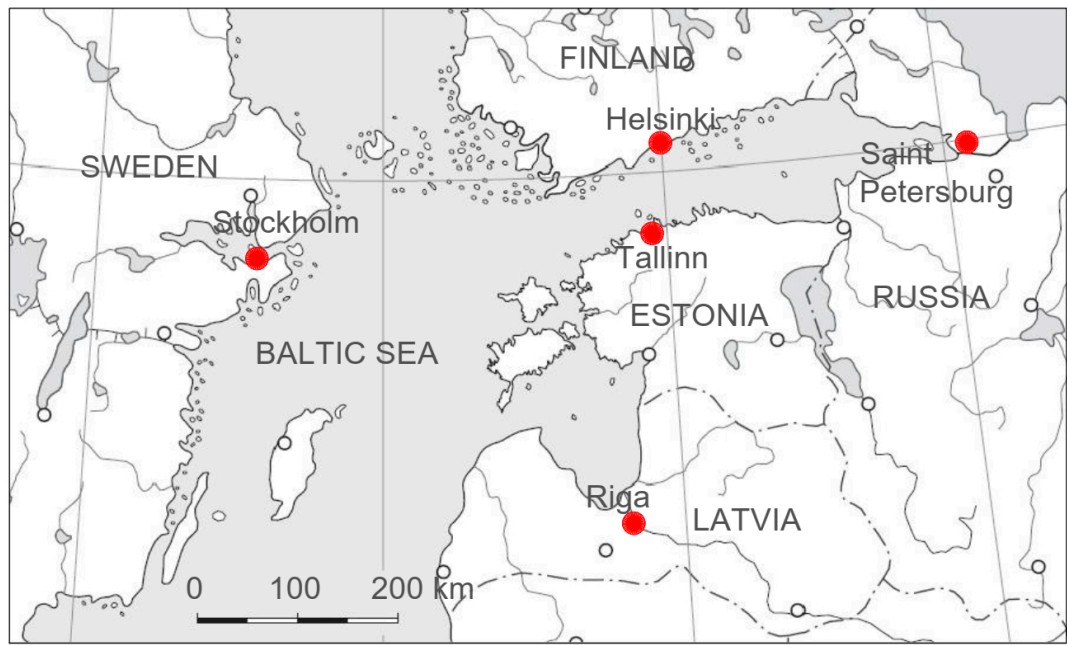

**Figure 1.** Map of the Central Baltic area.

The data of air temperatures that are used for the analysis are derived from public sources and are presented in Supplementary material (Table S1).

Meteorological observations in Stockholm started in 1754 and are some of the longest uninterrupted observation data in the world. The daily and monthly average harmonized measurement results of the Stockholm old astronomical observatory date back to 1756 [33,34]. The average monthly temperatures in Stockholm for the years 1756–2017 are obtained from the database [35].

In Tallinn, the air temperature has been measured at seven locations. Tarand et al. [14] found the location corrections, taking into account the effects of mesoclimate on the results of temperature measurements of meteo stations located on the territory of Tallinn. Tarand et al. used the location corrections and results from the earlier and later systematic meteorological observations at several observation points for the reconstruction of a time-series of average monthly air temperatures in Tallinn from 1756–2000, which is now known as the Tallinn-Maarjamäe observation point (59°27′ N; 24°48′ E). The results have been published in the monograph [14]. Currently, the air temperatures in Tallinn are measured at the Tallinn-Harku meteorological station of the Estonian Weather Service (59°24′ N; 24°36′ E). The Estonian Weather Service publishes the results of meteorological observations in yearbooks, the latest being with data from 2017 [36]. The data of Estonian weather observations are stored in the databases of Statistics Estonia (SE; http://www.stat.ee), which collects and stores data according to the international classifications and methods. At present, the meteorological station in Maarjamäe is no longer operative. With the use of parallel data of the air temperature measurements in Tallinn-Harku and Tallinn-Maarjamäe during 1992–2000, the air temperatures of Tallinn-Maarjamäe have been reduced to the Tallinn-Harku observation point on the basis of a mutual relationship between the average monthly air temperatures of these observation points. This provided an opportunity to reconstruct the series of average monthly temperatures in Tallinn-Harku from 1756–2017.

Air temperature measurements in Riga started in February 1795. There are 7 different measuring points, of which 6 are downtown or in the suburbs. A new meteorological station of the Meteorological Institute was opened in 1950, outside of the city center. Tarand et al. used the location corrections and results of the systematic meteorological observations in the nearby cities for the reconstruction of a time-series of average monthly air temperatures in Riga for the years 1796–1999 [14]. The average monthly temperatures of the Latvian Environment, Geology and Meteorology Centre in Riga, for the years 2004–2017 are obtained from the database of the Central Statistical Bureau of Latvia [37].

The data on the average monthly air temperatures in Riga for the years 2000–2003 are found on the basis of the data from the Berkeley Earth database (average monthly air temperatures in 1951–1980, and the anomalies in relation to these averages) [38].

The data on the average monthly air temperatures in Helsinki (Helsinki-Kaisaniemi) for the years 1829–1958 are obtained from the Berkeley database [39]. The data on the air temperatures for the years 1959–2017 are obtained from the database of the Finnish Meteorological Service [40].

The data on the air temperatures of the Saint Petersburg weather station from January 1834 to September 2016 are obtained from the Russian air temperature database [41], where the data has been specifically described [42]. The average monthly air temperatures in Saint Petersburg from October 2016 to December 2017 are calculated on the basis of the fixed term measurement results that are published on the Internet [43].

The location of the meteorological stations and the time series of the analyzed temperatures are shown in Table 1.

**Table 1.** Meteorological stations included in the study and periods of the time analyzed temperature data.

| Meteorological Station | Country | Coordinates | Period of Time |
|---|---|---|---|
| Stockholm | Sweden | 59°21′ N, 18°03′ E | 1756–2017 |
| Tallinn | Estonia | 59°24′ N; 24°36′ E | 1756–2017 |
| Riga | Latvia | 57°05′ N, 25°08′ E | 1796–2017 |
| Helsinki | Finland | 60°12′ N; 24°57′ E | 1829–2017 |
| Saint Petersburg | Russia | 59°58′ N, 30°18′ E | 1834–2017 |

The data from the temperature anomaly database of the Met Office Hadley Centre are also used for the assessment of changes in air temperatures [44].

The average annual air temperatures $t_y$ were found as the weighted average of the average monthly temperatures

$$t_y = \frac{\sum_i n_i t_i}{\sum_i n_i} \tag{1}$$

where, $i$ is the consecutive number of month, $n$ is the number of days per month, and $t$ is the average temperature of the month.

The average annual temperatures (1) depend on the locality and greatly fluctuate with time. Therefore, to compare the variability and trends of the temperatures of various localities, it is appropriate to examine the fluctuations in relation to the average of a certain period. The fluctuations/anomalies of annual average air temperatures are calculated in relation to the average temperatures in 1961–1990. Since the air temperature anomalies of the consecutive years are quite varying, we use a 10-year moving average of the temperature anomalies for observing the tendencies of changes.

The trends of changes in air temperature have been calculated by the method of least squares [45] for characterizing the changes in average air temperatures.

The meteorological conditions are formed due to a multitude of factors that make it difficult to determine a cause-and-effect relationship, which is possible in the case of a specific situation only. However, every specific meteorological situation may be regarded as one realization of every possible meteorological condition at a given location that has some probability of occurrence. In the course of time, given the large variability of air temperatures, statistical methods are used in this research for finding out the parameters that characterize the change in air temperature [45–47].

Due to their greater variability, the annual average air temperatures can be considered as random variables characterized by a distribution function. One question of interest is the frequency of occurrence of average annual and monthly air temperatures during some longer period and the possibility and pattern of change undergone by such distributions during longer periods (centuries). To study the frequency distribution of annual and monthly air temperatures, we have divided the

analyzed data into two groups based on centuries: the 18th century together with the 19th century and the 20th century together with the 21st century. The change of the mean value and variance of the distribution function of the air temperature for longer periods is a variable characteristic of climate change.

A visual assessment of the frequency distributions allows the assessment of the general nature of changes that have taken place; however, it gives no information on whether the changes hold any statistical relevance. The correspondence of the distribution function of monthly and annual air temperatures to the normal distribution is verified by Jarque-Bera and Smirnov-Kolmogorov tests [31]. For the normal distribution, the methods of verification of the statistical hypotheses dealing with random variables with normal distribution can be used. In this study, for the verification of hypotheses, attempts to prove a meaningful hypothesis (average air temperature has risen), rebutting the null hypothesis (air temperature has not risen) are carried out. For that purpose, a test statistic is calculated on the basis of the data of a sample, the theoretical distribution of which in the case of the validity of the null hypothesis is known. In case the value of the test statistic found is improbable compared to its theoretical distribution, the null hypothesis is deemed to be refuted, and the meaningful hypothesis proved.

In order to check the statistical relevance of the change of the average temperatures of various periods, Welch's *t*-test was used [45,47]. If the air temperature of a period has normal distribution, it is characterized by average temperature $\mu_1$ and the temperature dispersion $\sigma_1^2$. The average temperature for another period compared is $\mu_2$ and the temperature dispersion is $\sigma_2^2$. The real values of the average temperatures and dispersions are unknown to us. For one period $n_1$ and the other period $n_2$, we have the measured values of temperature; they can be regarded as a sample from the continuous distribution of temperatures. The *t*-test determines whether the difference between the average temperatures $\overline{x}_1$ and $\overline{x}_2$, obtained from the measured values, is statistically relevant.

Let us formulate a null hypothesis that the average temperatures of the periods compared are equal $H_0 : \mu_1 - \mu_2 \leq 0$. The alternative hypothesis is that the average temperatures have risen: $H_1 : \mu_1 - \mu_2 > 0$. For calculating the test statistic, we used the Equation (2)

$$t_0 = \frac{\overline{x}_1 - \overline{x}_2}{\sqrt{\frac{s_1{}^2}{n_1} + \frac{s_2{}^2}{n_2}}} \tag{2}$$

where $s_1$ and $s_2$ are standard deviations of the measured temperatures (samples).

The null hypothesis can be rejected if $t_0 > t_{\alpha,\nu}$. $t_{\alpha,\nu}$ is a value of Student's *t*-function or critical value of the test statistic corresponding to the significance level or error probability $\alpha$ and the number of degrees of freedom $\nu$. The number of degrees of freedom $\nu$ is calculated with:

$$\nu = \frac{\left(\frac{s_1{}^2}{n_1} + \frac{s_2{}^2}{n_2}\right)^2}{\frac{\left(s_1{}^2/n_1\right)^2}{n_1-1} + \frac{\left(s_2{}^2/n_2\right)^2}{n_2-1}} \tag{3}$$

If the empirical value of the test statistic is higher than the critical value, it can be concluded that the alternative hypothesis applies (average temperature has risen).

## 3. Results and Discussion

### 3.1. Changes in Average Annual Air Temperatures

First, the change of average annual air temperatures over a long period in the Central Baltic coastal area is of interest.

The average annual temperatures in the cities of the Central Baltic coastal area have been close, varying only by 1.7 °C as a long-term average. The average annual air temperatures of the entire history of measurement are the following: 5.8 °C in Stockholm, 4.9 °C in Tallinn, 6.3 °C in Riga, 5.2 °C

in Helsinki, and 4.6 °C in Saint Petersburg. Average air temperatures of different years vary quite a lot, fluctuating up to 6.5 °C. The average annual air temperatures have changed within the following ranges: 3.2 °C to 8.1 °C in Stockholm, 2.0 °C to 7.5 °C in Tallinn, 3.9 °C to 8.9 °C in Riga, 2.3 °C to 7.8 °C in Helsinki, and 1.3 °C to 7.7 °C in Saint Petersburg.

Figure 2 shows the fluctuations/anomalies of the average annual air temperatures in relation to the average temperatures of 1961–1990, comparing the variability and trends of air temperatures in different cities. The figure also shows a 10-year moving average of the temperature anomalies and a linear trend for the entire observed period. Despite the geographical proximity, the trends of changes in average annual air temperatures are quite different. So, the fastest rise of air temperature can be observed in Saint Petersburg (0.14 degrees per decade) and the lowest in Stockholm (0.04 degrees per decade). The average annual air temperature rise in Riga, Tallinn, and Helsinki has been 0.05, 0.06, and 0.09 degrees per decade, respectively.

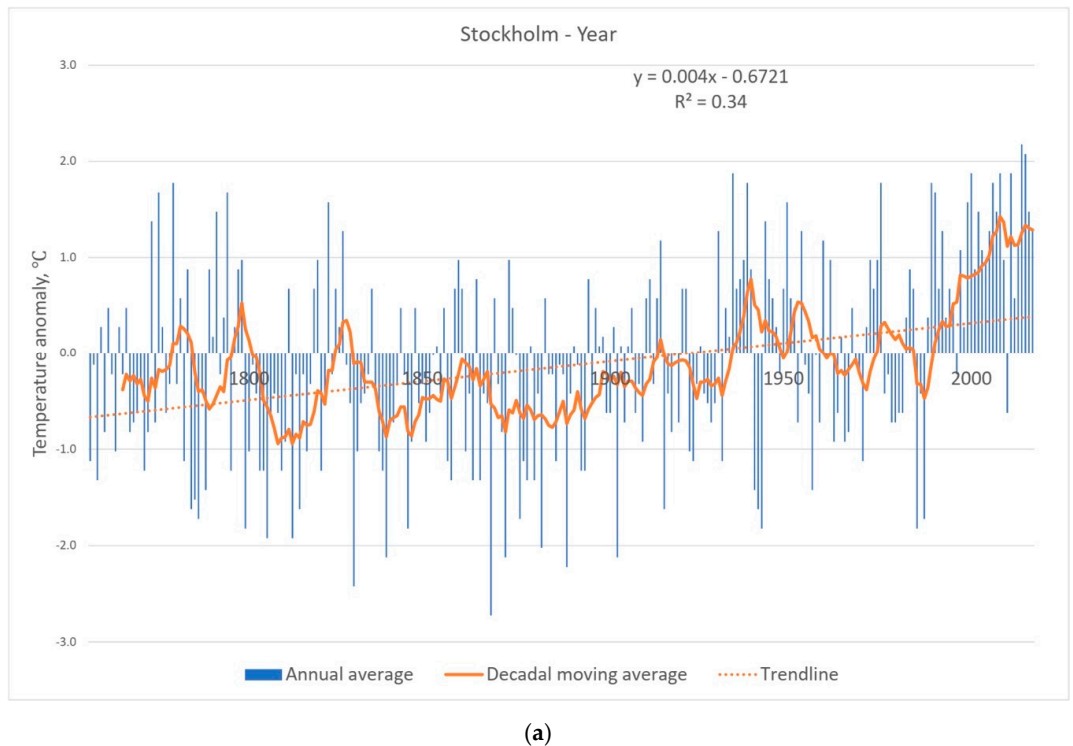

(**a**)

**Figure 2.** *Cont.*

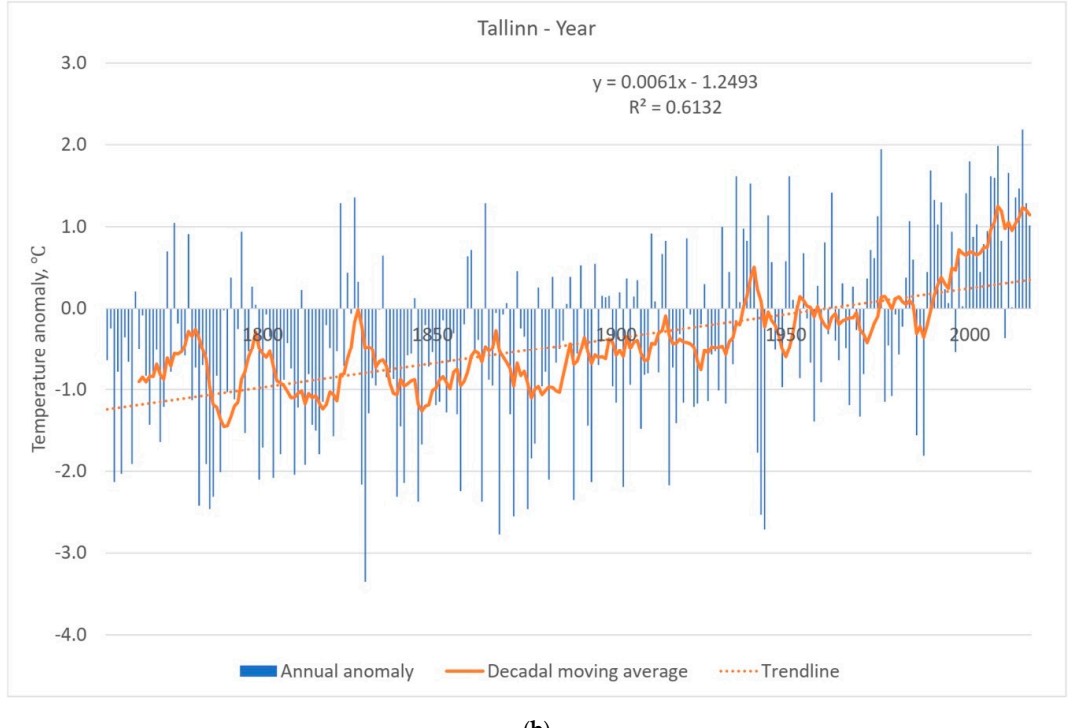

(**b**)

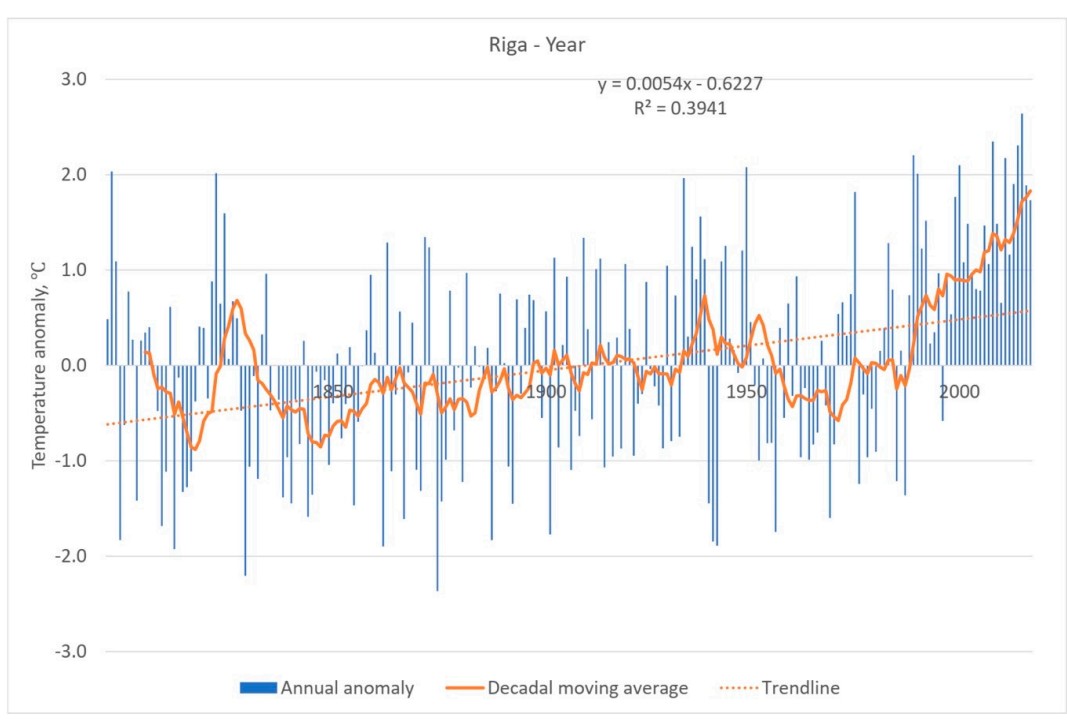

(**c**)

**Figure 2.** *Cont.*

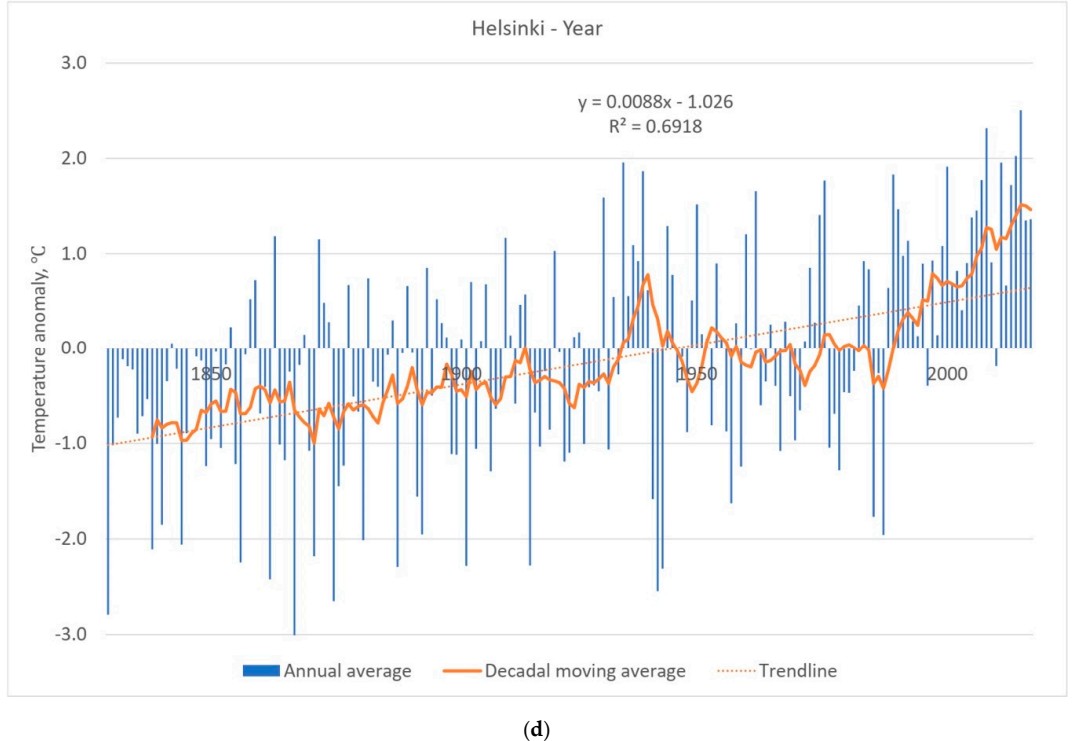

(**d**)

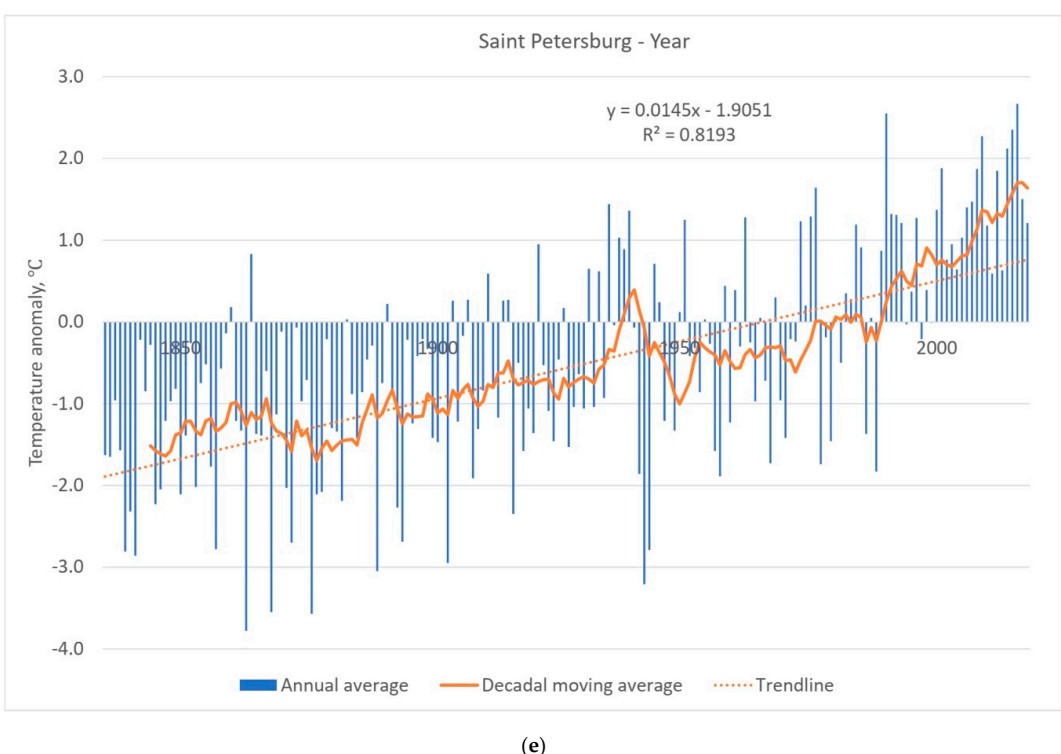

(**e**)

**Figure 2.** Average annual air temperature anomalies (difference in relation to the 1961–1990 average). (**a**) Stockholm; (**b**) Tallinn; (**c**) Riga; (**d**) Helsinki; (**e**) Saint Petersburg. The 10-year moving average is presented by the red line.

To allow a better comparison of the temporal changes in air temperature in different localities, the 10-year moving average of the anomalies of the average annual air temperatures in the Central Baltic cities is determined (Figure 3). The comparison of the global and northern hemispheric air temperature anomalies is based on the data of the of the Hadley HadCRUT4 data set [16] on the same temporal

scale. The average annual air temperature anomalies approximately follow the same temporal change pattern, with maximum 2 °C difference between them. It is noteworthy that the air temperatures in Saint Petersburg have risen the quickest in comparison to the other Central Baltic cities.

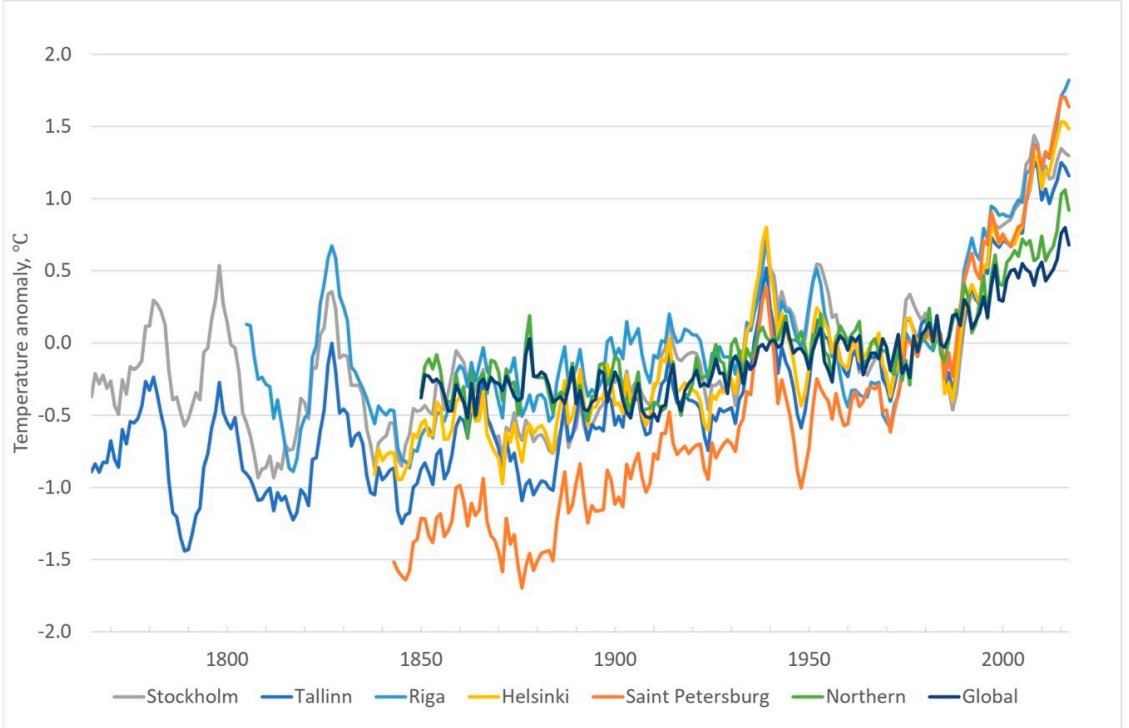

**Figure 3.** Ten-year moving averages of the average annual air temperature anomalies (difference in relation to the 1961–1990 average): Stockholm; Tallinn; Riga; Helsinki; Saint Petersburg. The anomalies of global and northern hemisphere air temperature are presented for comparison from the Met Office Hadley Centre database [44].

Changes in the air temperatures and variations in different periods are analyzed by using a long time-series of average air temperatures of the cities of the Central Baltic coastal area. Figure 4 shows the trends of the average annual air temperatures of the cities of the Central Baltic coastal area in different periods of time. It is evident that the rise of the average annual air temperature has been faster in the Central Baltic coastal area and in the northern hemisphere than it has been globally, especially during the recent shorter periods when the rise of the air temperature reached 0.5 °C per decade. The rapid increase of the average temperature is an important peculiarity of the Central Baltic region compared with other regions of Europe. Thus, trend per decade of the average annual air temperature of the Central England area during the period 1980–2017 is 0.27 °C [48] and trend per decade of the average annual air temperature of the Bosnia and Herzegovina is 0.32 °C [11]. The temperature changes for a short period may not be extrapolated; however, it can be done for longer periods as shown in Figures 2 and 3, where several periods have existed for a longer time during which the average annual air temperature has suddenly risen and then fallen.

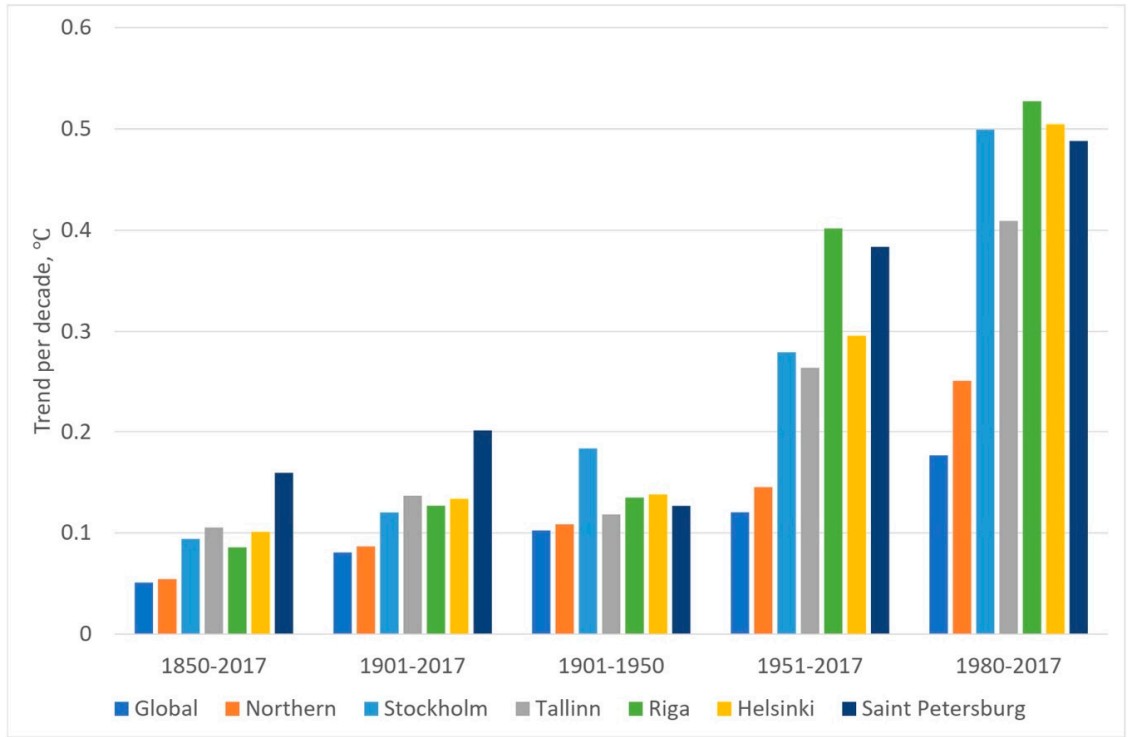

**Figure 4.** Trends of average annual air temperatures for ten years in different periods.

An analysis of the time series of temperatures does not make it possible to identify the causes for these changes. One possible factor influencing air temperature growth may be the additional heat flow to the atmosphere associated with urbanization. In simplified terms, the extent of urbanization can be characterized by population data. There is no or very little relationship between the rapid increase of the temperature of the cities of the Central Baltic area with urbanization. The correlation analysis shows that at confidence level 0.05 there is no relationship between air temperature and population data in the case of Stockholm, Tallinn, Riga, and Helsinki. In the case of Saint Petersburg, there is a medium correlative relationship between the air temperature and the population data (coefficient of correlation 0.52).

In this research, the air temperature is regarded as a random variable. Therefore, the frequency distribution of air temperatures is of interest. The study of the frequency distribution of the air temperatures of different periods enables the assessment of the tendencies of changes of air temperatures and the statistical reliability of the changes. Figure 5 shows the frequency distributions of the average annual air temperatures in the Central Baltic cities in different periods of time. For all the cities, the shift of the frequency curves is set towards the rise of temperatures. Jarque-Bera and Smirnov-Kolmogorov tests were used (significance level 0.05) to confirm that the frequency distributions of the average annual air temperatures correspond to the normal distribution. Hence, we can use Welch's *t*-test to assess the statistical reliability of the changes. At the significance level of 0.05, the critical value of the test statistic is 1.65, and the empirical values are as follows: Tallinn—6.83, Stockholm—5.25, Helsinki —4.94, Riga—3.54, and Saint Petersburg—9.60. Therefore, at the significance level of 0.05, the average annual air temperature during years 1901–2017 in comparison with the previous period of time has risen in all the observed cities.

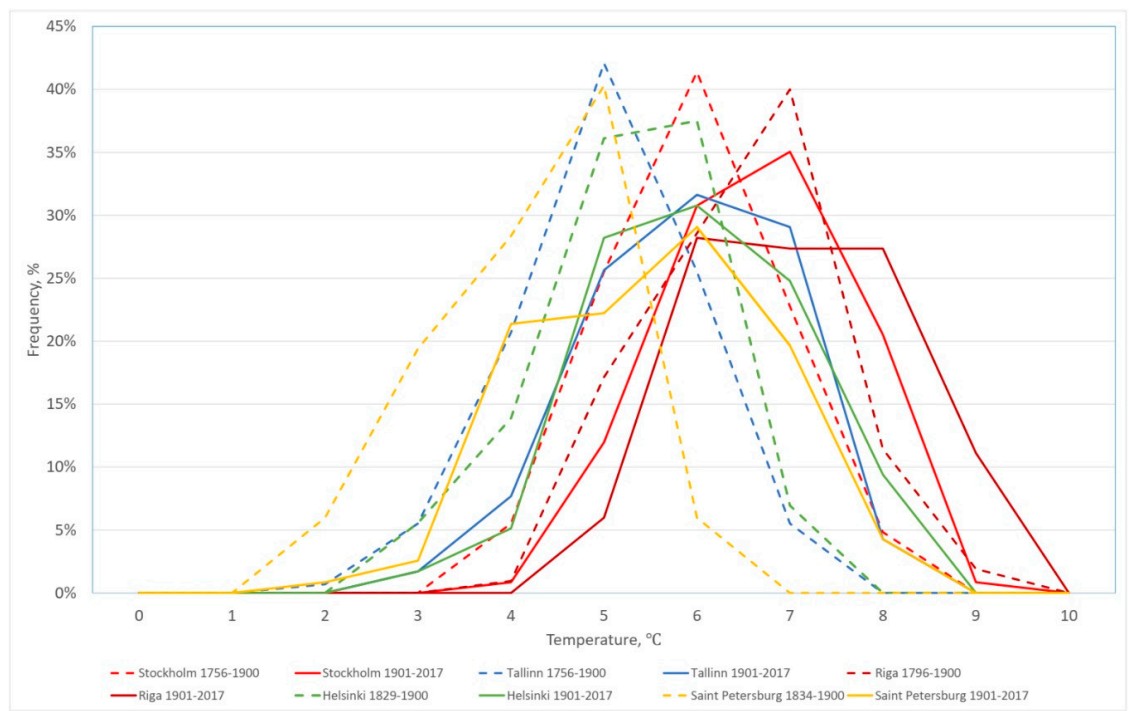

**Figure 5.** Frequency distribution of the average annual air temperatures in different periods.

*3.2. Changes in Average Monthly Air Temperatures*

The rise of average annual temperatures means that the average monthly temperatures have changed (risen) over time. Therefore, the changes in the average monthly air temperatures of different months should be ascertained.

Statistical indicators (median, average, standard deviation, minimum value, and maximum-value) of the average air temperatures in various periods in the Central Baltic cities are presented in Supplementary material (Table S2). The average annual temperatures in these cities do not differ much. The warmest city of the region is Riga where the average annual air temperatures have changed by 6.0 °C to 6.6 °C in centuries and the coolest is Saint Petersburg where the average annual air temperatures have risen by 3.8 °C to 5.0 °C.

The average monthly air temperature in the observed cities has risen during the 20th and the beginning of 21st century as follows: 0.94 °C in Stockholm, 1.14 °C in Tallinn, 0.78 °C in Riga, 1.02 °C in Helsinki, and 1.41 °C in Saint Petersburg. The median of the average monthly air temperature has risen during the century as follows: 1.07 °C in Stockholm, 1.14 °C in Tallinn, 0.76 °C in Riga, 1.13 °C in Helsinki, and 1.54 °C in Saint Petersburg.

Table S2 shows that the changes in the air temperatures in the observed periods have been different in different months and in different cities. For example, the average monthly temperature in different centuries has risen from −4.4 °C to −3.0 °C in Stockholm, 6.8 °C to −4.6 °C in Tallinn, −5.0 °C to −4.1 °C in Riga, −5.8 °C to −4.8 °C in Helsinki, and −8.5 °C to −7.0 °C in Saint Petersburg. In June, the rise of air temperature was smaller in Tallinn (from 13.5 °C to 13.9 °C) and Saint Petersburg (from 14.5 °C to 15.3 °C), remained unchanged in Helsinki (14.4 °C) and Riga (15.5 °C) and even dropped from 14.6 °C to 14.1 °C in Stockholm.

In order to assess the similarity between the temperature regimes of the observed cities, we will examine the frequency distributions of the average air temperatures of all the months throughout the entire history of measurement. The average monthly air temperatures vary to a great extent within a year, changing from −19.5 °C (Saint Petersburg, February) up to 24.4 °C (Saint Petersburg, July). Figure 6 shows the frequency distribution of the occurrence of all the average monthly air temperatures in Stockholm (1756–2017), Tallinn (1756–2017), Riga (1796–2017), Helsinki (1829–2017), and Saint

Petersburg (1834–2017). The percentages are calculated for 1 °C intervals. It appears that the frequency distributions of the average monthly air temperatures practically similar. This means that on average there are similar temperature conditions in the Central Baltic coastal area. However, Saint Petersburg differs a little due to lower average monthly air temperatures.

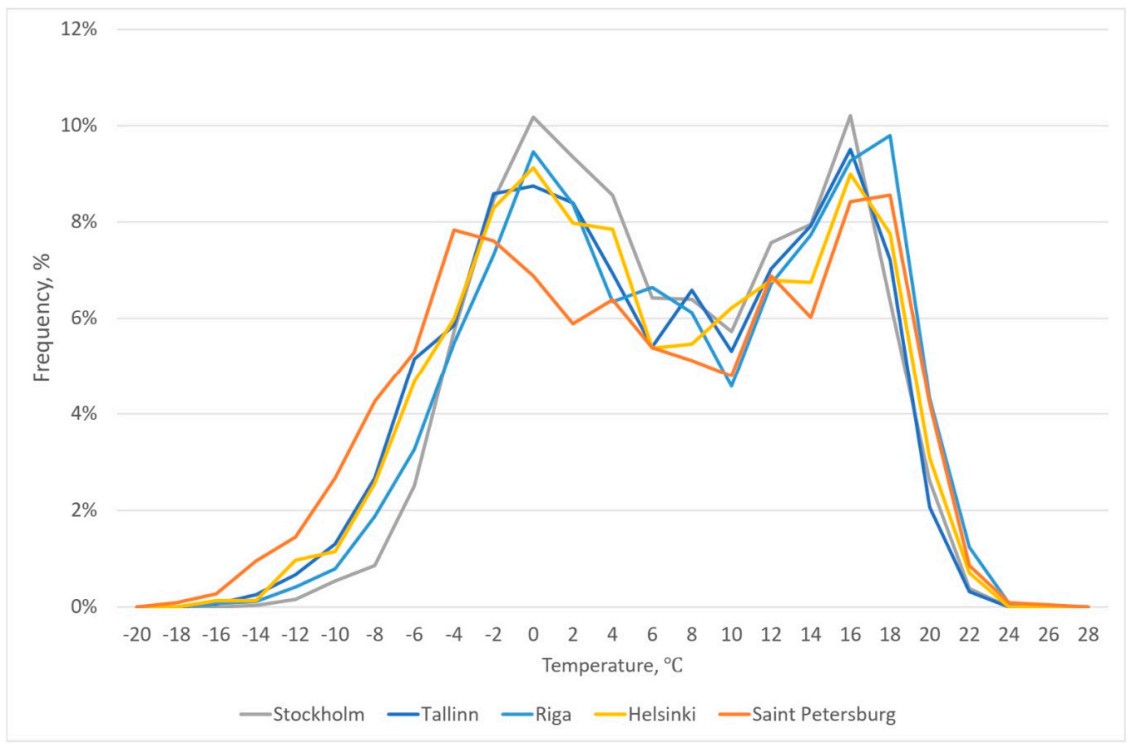

**Figure 6.** Frequency distribution of the average air temperatures of all months.

The standard deviation of the average monthly air temperatures changes from 1.4 °C (Tallinn and Stockholm—September) up to 3.9 °C (Saint Petersburg–January and February). Therefore, the variability of the air temperatures in different months is worthy of closer examination.

Due to their greater variability, the monthly average air temperatures can be considered as random variables, characterized by a distribution function. The average air temperature values occur with greater probability and are more extreme with a smaller probability.

One problem of interest is the frequency of occurrence of the average monthly air temperatures during longer periods and whether and how these distributions have changed during longer periods (centuries). In order to find out the nature of the air temperature changes, we will observe the frequency distributions of all average monthly air temperatures in the cities of the Central Baltic coastal area in different periods.

Figure 7 shows the frequency distribution of all the average monthly air temperatures in Stockholm, Tallinn, Riga, Helsinki, and Saint Petersburg in different periods of time. It is noteworthy that the changes are different in different months.

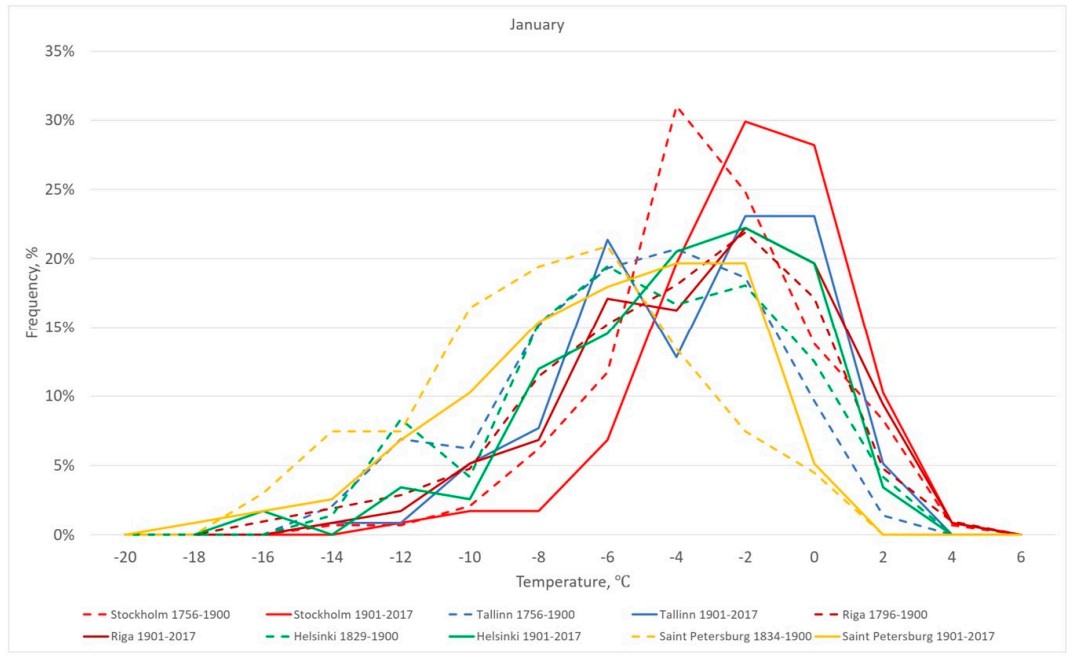

(**a**)

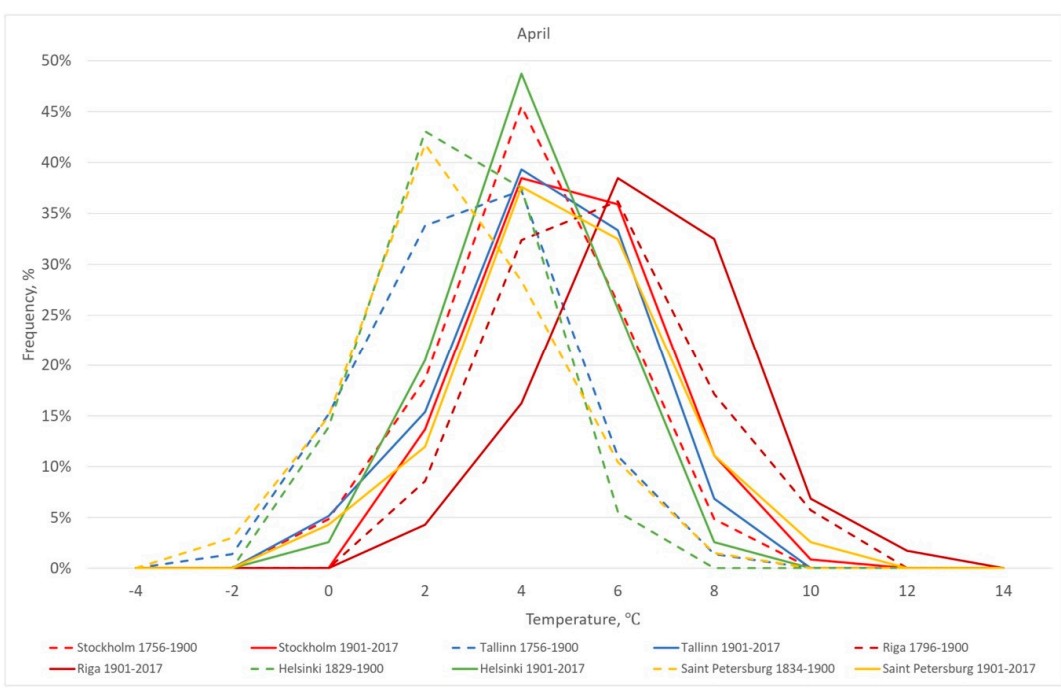

(**b**)

**Figure 7.** *Cont.*

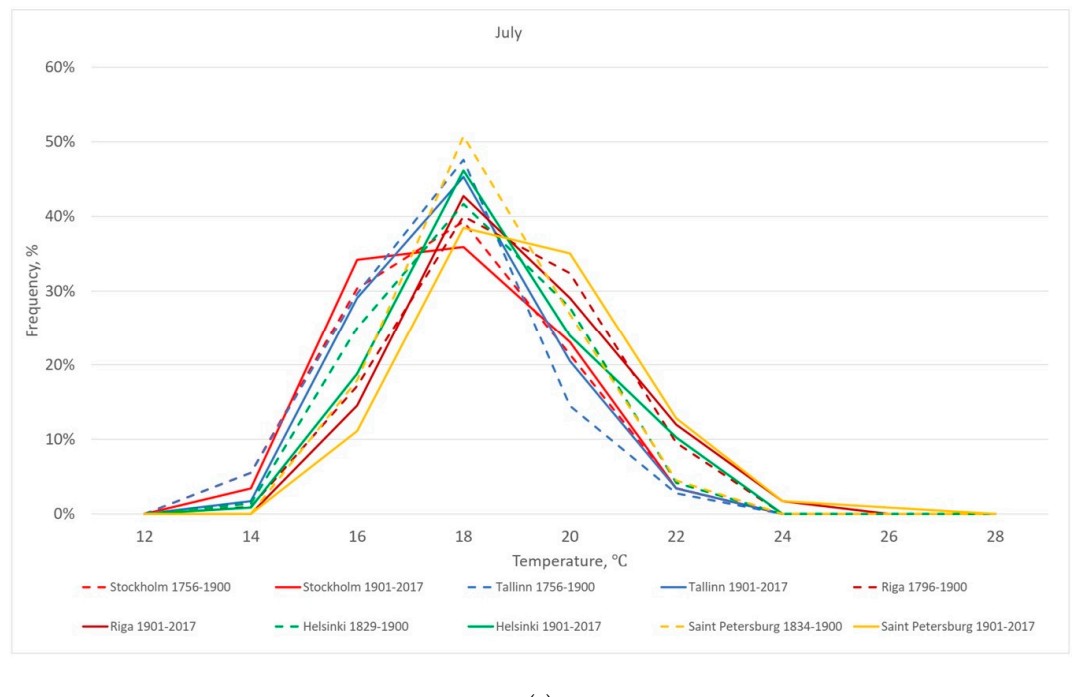

(**c**)

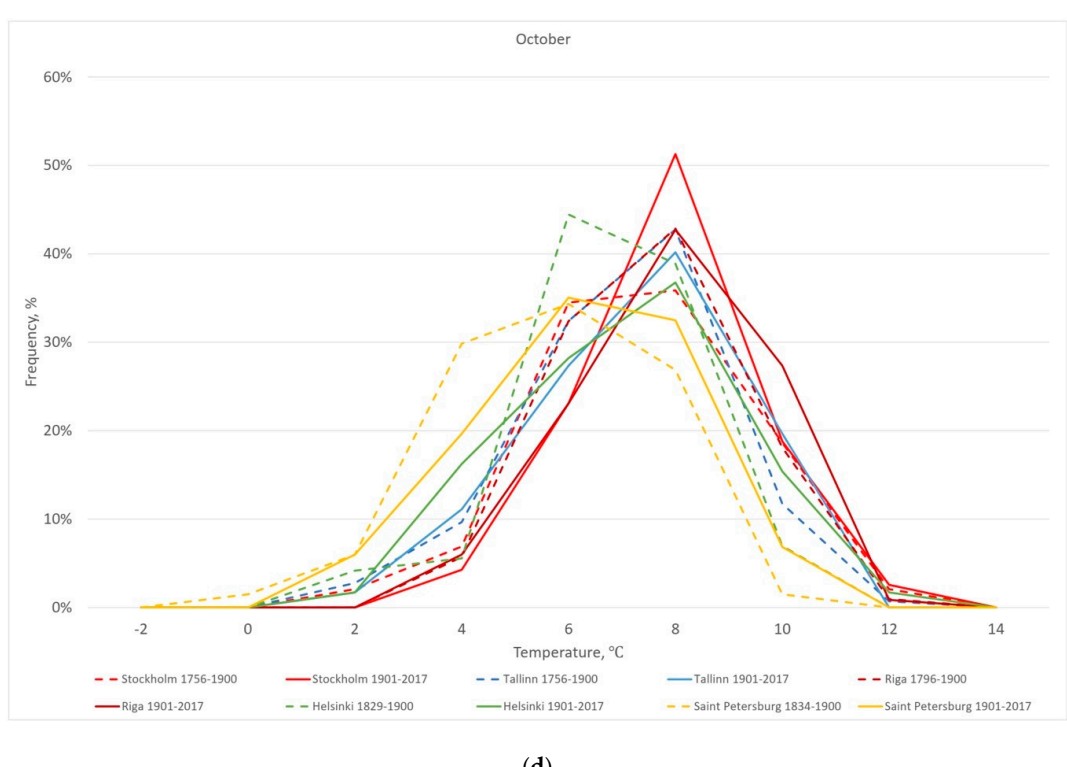

(**d**)

**Figure 7.** Frequency distributions of the average monthly air temperatures in different periods. (**a**) January; (**b**) April; (**c**) July; (**d**) October.

In January the temperatures are warmest in Stockholm, where the maximum frequency distribution of the air temperature shifted approximately 2 °C to the positive direction within a century, but this was not the highest rise of January temperature. The most common air temperature in Saint Petersburg has risen by 4 °C within a century. At the same time, there have been in Saint Petersburg colder Januaries in the last century than in the century before. In April, the highest air

temperatures are in Riga. The frequency distributions of temperatures have shifted 2 °C, on an average, in the positive direction. In July, the frequency distributions of the average air temperatures have remained practically the same within a century, only a slight rise of the air temperatures of Saint Petersburg can be observed. In October, there was rise of the air temperatures in Saint Petersburg.

A visual assessment of the frequency distributions allows the analysis of the general nature of changes that have taken place; however, it gives no information on whether the changes hold any statistical relevance. Applying the Jarque-Bera test, the frequency distributions of the average monthly temperatures do not conform to the normal distribution at the significance level of 0.05, but the non-conformities are not very big. At the same time, based on the Smirnov-Kolmogorov test, the distribution of the average monthly air temperatures may be regarded as conforming to the normal distribution at the significance level of 0.05. Thus, we can use the methods of verification of statistical hypotheses that deal with random variables with normal distribution when we study the changes of average monthly air temperatures.

In order to check the statistical relevance of the change of the average temperatures of various periods, we use Welch's *t*-test. The results show (Figure 8) that the average annual air temperatures in the whole Central Baltic coastal area have risen in the 20th century (together with the beginning of the 21st century) at the significance level 0.05 in comparison with the 19th century. But the changes in the 20th century in comparison with the 19th century have not occurred in the same way in different months. Only for Saint Petersburg, it can be said that the average air temperature has risen in all months. The same cannot be confirmed, however, about the average air temperature of Riga in February and from June to September. There has been no significant rise of the average air temperature from June to September in Stockholm, from June to October in Helsinki, and in August and September in Tallinn within the century. The result of the statistical analysis shows that in the cities of the Central Baltic coastal area, except Saint Petersburg, the air temperature has risen in some months, while in some months there has not been any rise of the air temperature.

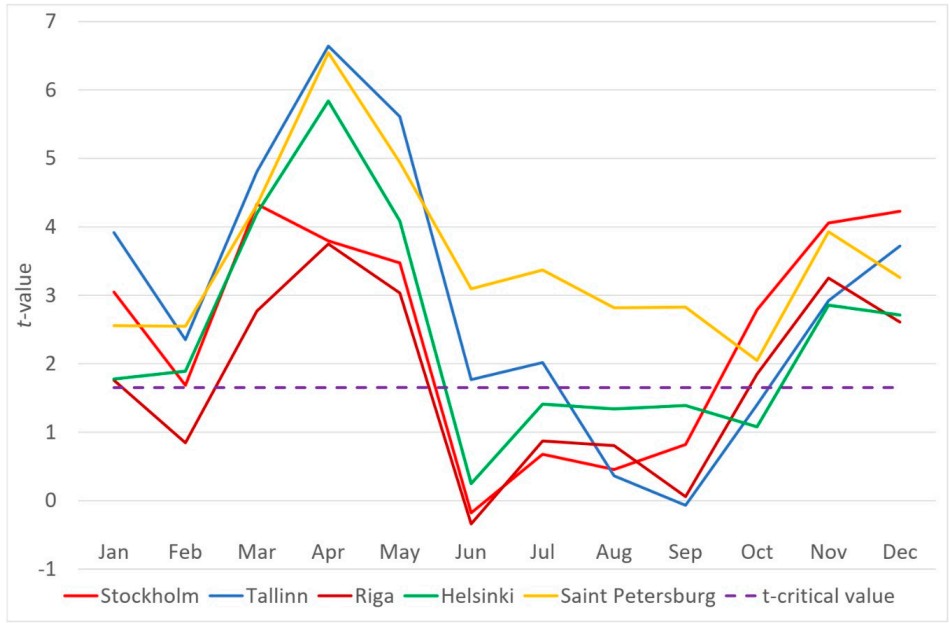

**Figure 8.** Verification of the hypothesis of rise of the air temperature at the significance level of 0.05. The null hypothesis (average temperature has not risen) is valid in the critical area of the parameter (below the dashed line). Comparison periods of the average temperatures: Stockholm: the period of 1901–2017 in relation to the period of 1756–1900; Tallinn: the period of 1901–2017 in relation to the period of 1756–1900; Riga: the period of 1901–2017 in relation to the period of 1796–1900; Helsinki: the period of 1901–2017 in relation to the period of 1829–1900; Saint Petersburg: the period of 1901–2017 in relation to the period of 1834–1900.

The results make the study of temporal change dynamics of average monthly air temperatures of different months important. We observe in more detail the anomalies of the average air temperature of four months (January, April, July, and October) (difference in relation to the average air temperature in 1961–1990).

Figure 9 shows ten-year moving averages of the January air temperature anomalies (difference in relation to the 1961–1990 average): Stockholm; Tallinn; Riga; Helsinki; Saint Petersburg. The time-series of anomalies of January air temperature of the cities of the Central Baltic coastal area with a 10-year moving average and linear trend line are presented in the Supplementary material (Figure S1a–e). The smallest rise of the average temperature in January has been in Stockholm (a rise of 0.09 °C per decade), the highest in Saint Petersburg (0.17 °C per decade).

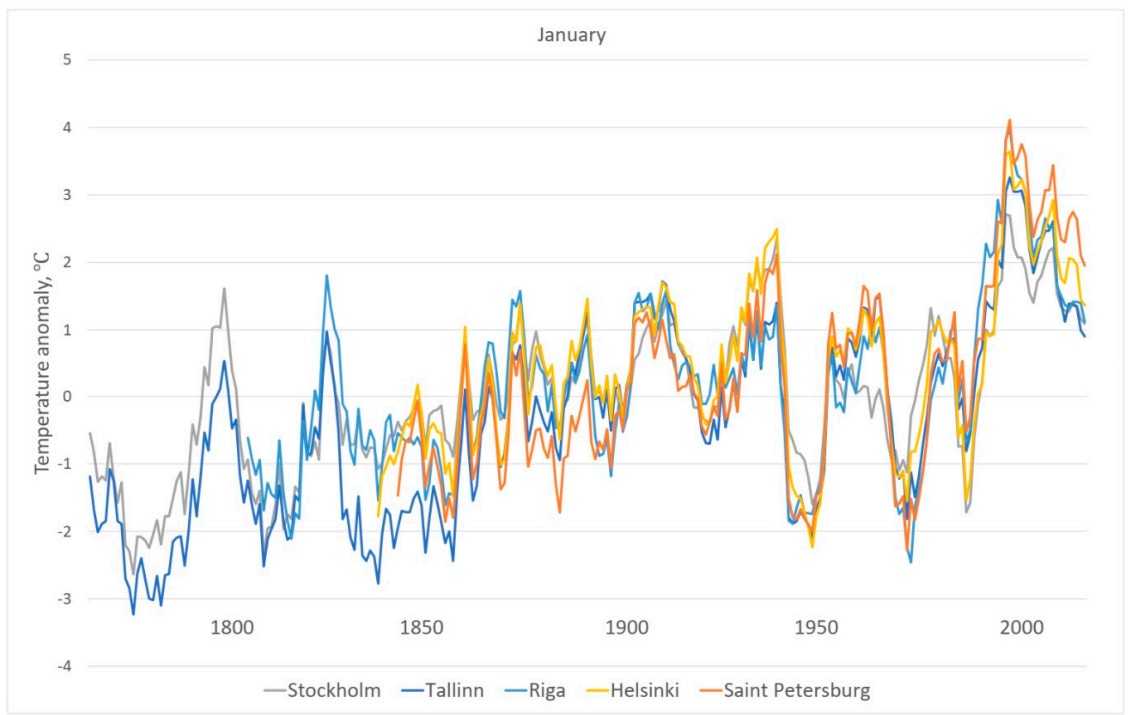

**Figure 9.** Ten-year moving averages of the January air temperature anomalies (difference in relation to the 1961–1990 average) of the cities of the Central Baltic coastal area.

Figure 10 shows ten-year moving averages of the April air temperature anomalies (difference in relation to the 1961–1990 average): Stockholm; Tallinn; Riga; Helsinki; Saint Petersburg. The time-series of anomalies of April air temperature of the cities of the Central Baltic coastal area with a 10-year moving average and linear trend line are presented in Supplementary material (Figure S2a–e). The smallest rise of the average temperature in April has been in Stockholm (rise of 0.05 °C per decade), and the biggest in Saint Petersburg (0.23 °C per decade).

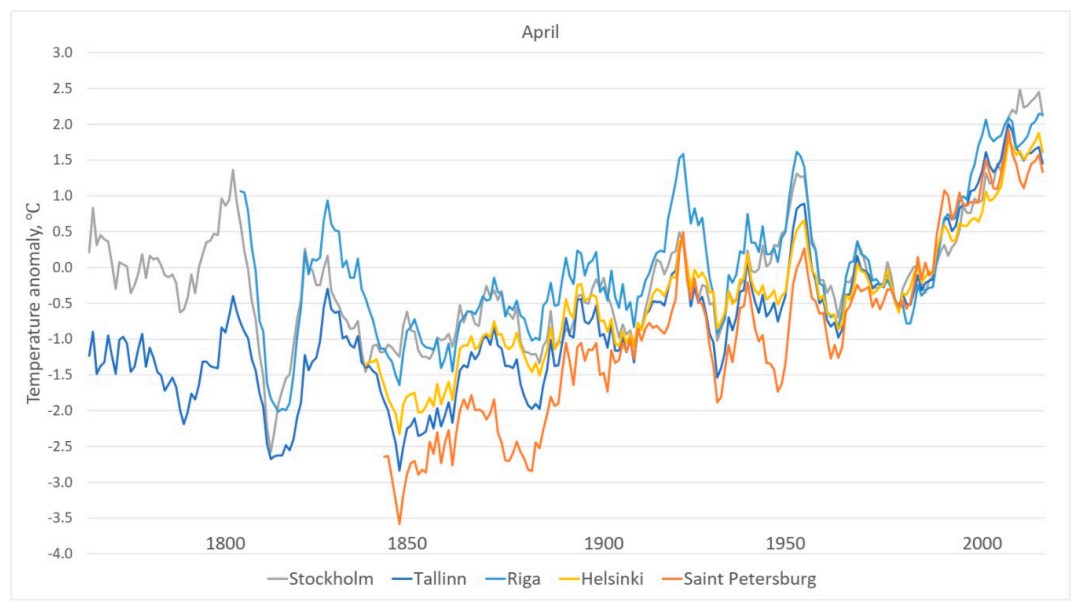

**Figure 10.** Ten-year moving averages of the April air temperature anomalies (difference in relation to the 1961–1990 average) of the cities of the Central Baltic coastal area.

Figure 11 shows ten-year moving averages of the July air temperature anomalies (difference in relation to the 1961–1990 average): Stockholm; Tallinn; Riga; Helsinki; Saint Petersburg. The time-series of anomalies of July air temperature of the cities of the Central Baltic coastal area with a 10-year moving average and linear trend line are presented in Supplementary material (Figure S3a–e). The average temperature in July has not risen at all in Stockholm (trend 0.00 °C per decade), but the rise was highest in Saint Petersburg (rise of 0.23 °C per decade).

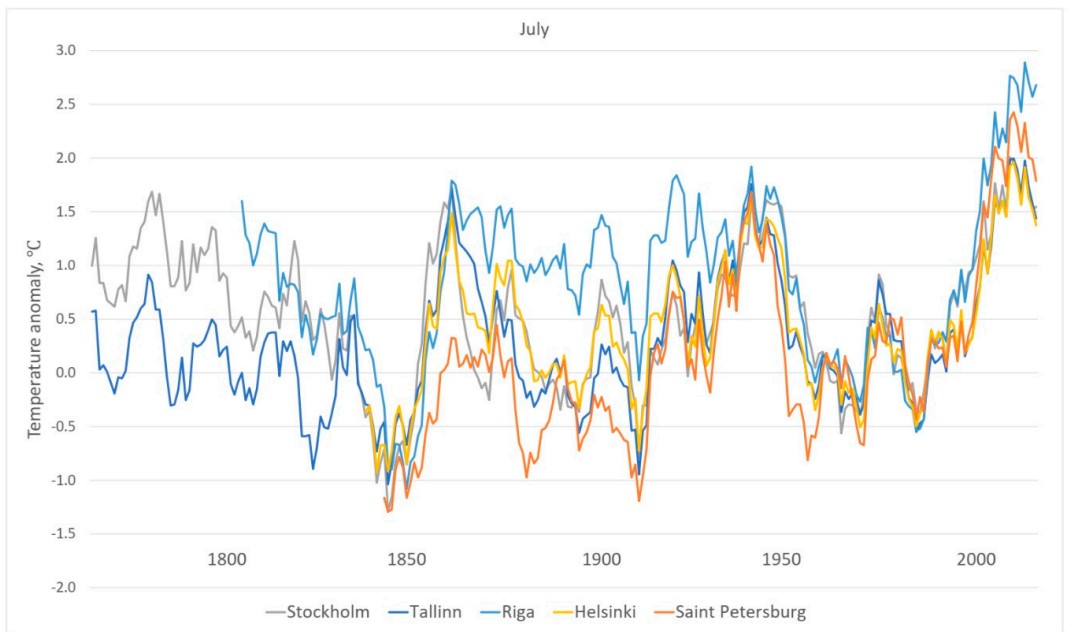

**Figure 11.** Ten-year moving averages of the July air temperature anomalies (difference in relation to the 1961–1990 average) of the cities of the Central Baltic coastal area.

Figure 12 shows the time-series of the air temperature anomalies of the cities of the Central Baltic coastal area in October. The time-series of anomalies of October air temperature of the cities of the Central Baltic coastal area with a 10-year moving average and linear trend line are presented

in Supplementary material (Figure S4a–e). The smallest rise of the average temperature in October has been in Tallinn (rise of 0.02 °C per decade), but the highest in Saint Petersburg (rise of 0.09 °C per decade).

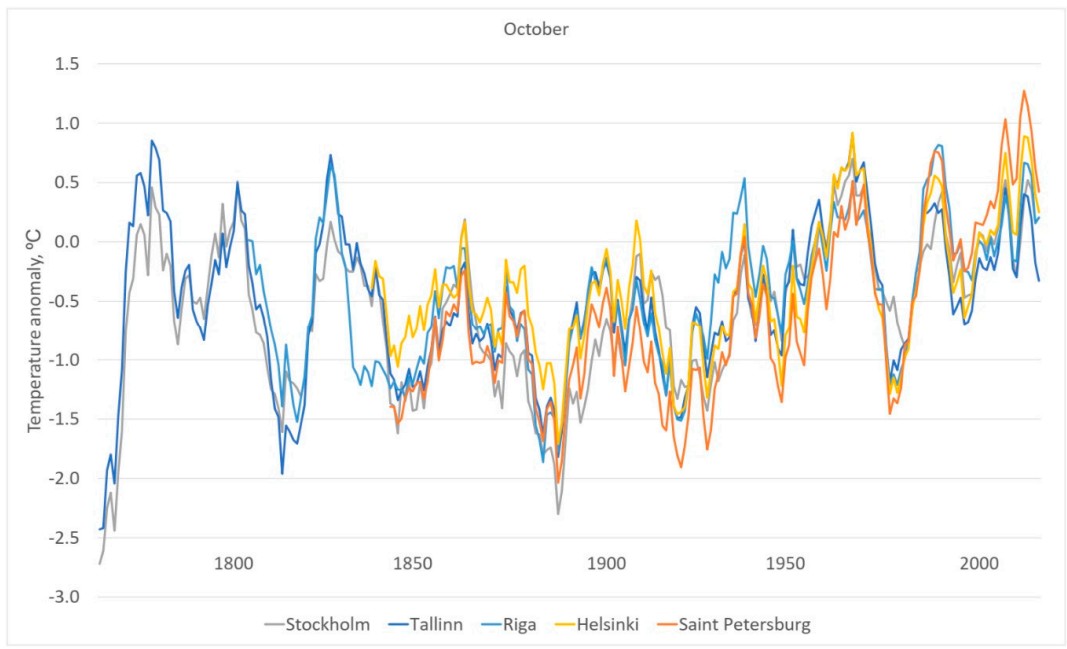

**Figure 12.** Ten-year moving averages of the October air temperature anomalies (difference in relation to the 1961–1990 average) of the cities of the Central Baltic coastal area.

After studying the long-term trends of the air temperatures of all months, it becomes evident that the trends have a certain annual course. Figure 13 shows the annual course of the average monthly trends of air temperatures in the Central Baltic coastal area (°C per decade).

Per decade, the air temperature rise has reached a maximum in March and April, reaching 0.09 °C (Stockholm, Tallinn) up to 0.23 °C (Saint Petersburg). From June to September, the rise of air temperature is considerably lower, remaining below 0.04 °C per ten years. The rise of air temperature in February is noteworthy. Thus, the air temperature changes are practically absent during the summer period and are smaller in mid-winter. Significant changes have occurred in autumn and in spring. Such annual course of the trends in the average monthly air temperatures is an important peculiarity of the Central Baltic region compared with other regions of Northern Hemisphere. For example, in Bosnia and Herzegovina, the minimum temperature trend of the monthly average air temperature is in February and in September and the maximum is in January and in August [11]. In Junagadh (India), the minimum trend of the monthly average air temperature is in September and the maximum is in February, in June and in November [12]. In Taiz City (Yemen), the minimum trend of the monthly average air temperature is in June and the maximum is in May to July [13]. In Central England the minimum trend of the monthly average air temperature is in September and the maximum is in January, in March, and in November [48].

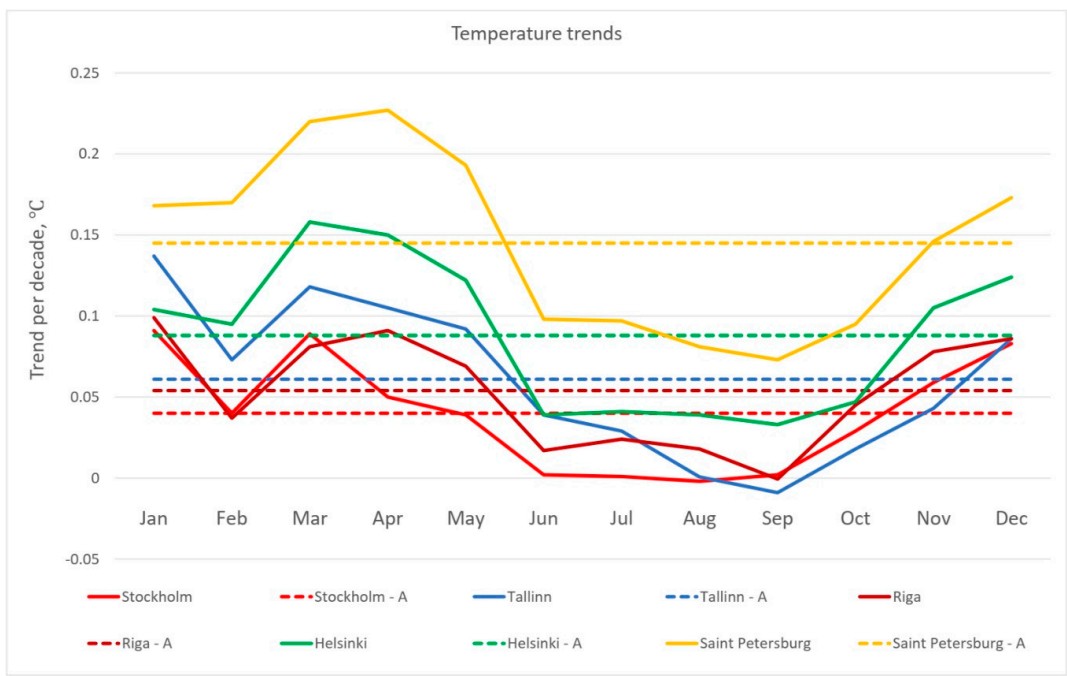

**Figure 13.** Annual course of the trends in the average monthly air temperatures (°C per decade). The figure also shows the values of the trends of the average annual air temperatures A (°C per decade) for comparison.

It is interesting that the rise of the air temperature is faster in localities that are closer to the Arctic region (Saint Petersburg, Helsinki) and smaller in regions that are more to the South and separated from the Arctic region by the Baltic Sea (Stockholm, Tallinn, Riga).

## 4. Conclusions

Differences in the air temperatures of the cities of the Central Baltic coastal area are relatively small; the long-term average annual air temperatures differ slightly. The long-term average temperatures differ with 1.7 °C; the frequency distributions of the average air temperatures of all months are practically the same.

Near the ground, air temperatures in the Central Baltic coastal area have changed twice as fast as in the northern hemisphere as on earth as a whole. During 1850–2017, the 10-year average rise of the air temperature has been 0.09 °C in Stockholm, 0.11 °C in Tallinn, 0.09 °C in Riga, 0.10 °C in Helsinki, and 0.16 °C in Saint Petersburg. At the same time the 10-year rise of the air temperature has been 0.05 °C globally as well as in the northern hemisphere, or the rise has been twice slower. The same ratio also exists at present. During 1980–2017, the 10-year average rise of the air temperature has been 0.50 °C in Stockholm, 0.41 °C in Tallinn, 0.53 °C in Riga, 0.50 °C in Helsinki, and 0.49 °C in Saint Petersburg. At the same time the 10-year rise of the air temperature has been 0.18 °C globally and 0.25 °C in the northern hemisphere.

It was found using statistical methods that at the significance level of 0.05, the average annual air temperature has risen in all the cities of the Central Baltic coastal area in the period of 1901–2017 in comparison to the previous period.

The result of the statistical analysis shows that in the cities of the Central Baltic coastal area, except Saint Petersburg, the air temperature has risen in some months while there was no change in some months, within a century. The average air temperature has risen in Saint Petersburg in all months within a century. In Riga, the average air temperature has not risen in February and from June to September. The average air temperature has not risen from June to September in Stockholm, from June to October in Helsinki, and in August and September in Tallinn. During the last few decades, the rise

of the average annual air temperature in Saint Petersburg has remained lower than its neighboring cities Helsinki, Tallinn, and Stockholm, confirming the veracity of our version.

The average monthly air temperatures have not changed during a longer period of time. The trends of the average monthly air temperature have a certain annual course. The rate of rise of the air temperature is slightly different in different cities, but the changes are similar in all cities. The air temperature has risen most in March and April, reaching 0.09 °C (Stockholm, Tallinn) up to 0.23 °C (Saint Petersburg) per ten years. From June to September, the rise of the air temperature is considerably lower, remaining below 0.04 °C per ten years. The rise of the air temperature is relatively smaller in February also. Thus, the air temperature changes are practically absent during the summer period and are smaller in mid-winter. Significant changes have occurred in autumn and in spring.

Changes in the air temperature in the cities of the Central Baltic coastal area have not occurred evenly, neither on the spatial nor the temporal scale. The rise of the air temperature is faster in localities that are closer to the Arctic region (Saint Petersburg, Helsinki) and smaller in regions that are more to the South and separated from the Arctic region by the Baltic Sea (Stockholm, Tallinn, Riga).

**Supplementary Materials:** The following are available online at http://www.mdpi.com/2225-1154/7/2/22/s1, Figure S1: Ten-year moving averages and a linear trend of the average January air temperature anomalies (difference in relation to the 1961–1990 average): (a) Stockholm; (b) Tallinn; (c) Riga; (d) Helsinki; (e) Saint Petersburg, Figure S2: Ten-year moving averages and a linear trend of the average April air temperature anomalies (difference in relation to the 1961–1990 average): (a) Stockholm; (b) Tallinn; (c) Riga; (d) Helsinki; (e) Saint Petersburg, Figure S3: Ten-year moving averages and a linear trend of the average July air temperature anomalies (difference in relation to the 1961–1990 average): (a) Stockholm; (b) Tallinn; (c) Riga; (d) Helsinki; (e) Saint Petersburg, Figure S4: Ten-year moving averages and a linear trend of the average October air temperature anomalies (difference in relation to the 1961–1990 average): (a) Stockholm; (b) Tallinn; (c) Riga; (d) Helsinki; (e) Saint Petersburg, Table S1: Monthly average air temperatures (°C): (a) Stockholm; (b) Tallinn; (c) Riga; (d) Helsinki; (e) Saint Petersburg, Table S2: Statistical indicators of the average air temperatures (°C) during different time-periods: (a) Stockholm; (b) Tallinn; (c) Riga; (d) Helsinki; (e) Saint Petersburg.

**Funding:** This research received no external funding.

**Conflicts of Interest:** The author declares no conflict of interest.

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
