# Peer review of "Peculiarities of Long-Term Changes in Air Temperatures Near the Ground Surface in the Central Baltic Coastal Area"

_climate, doi:10.3390/cli7020022_

Reviewer 1 Report

Brief summary

This is a classical climate change paper. The aim is to detect, characterize and compare the long-term trend in the average surface air temperature in five locations in the Central Baltic coastal area. The author benefits from having long time series covering approximately the same period. The main contributions include the assessment of trends in annual and monthly time series, the comparison between results obtained in the different locations. However, the manuscript has several limitations and cannot be accepted in its current state.

My overall Recommendation is “Reconsider after Major Revisions” which means that he acceptance of the manuscript would depend on the revisions. The author needs to provide a point by point response or provide a rebuttal if some of the reviewer’s comments cannot be revised. Usually, only one round of major revisions is allowed. Authors will be asked to resubmit the revised paper within ten days and the revised version will be returned to the reviewer for further comments.

General comments/suggestions to the authors,

The manuscript was rated in different aspects

Originality/Novelty: The research questions are not entirely original (in the sense that the similar questions have been raised in other studies, even by the author). The results are in line with the present general knowledge (of global warming). There are other studies on this topic in the region. Please see the BACC project (BALTEX Assessment of Climate Change for the Baltic Sea Basin) [Team, B. A. (2008). Assessment of climate change for the Baltic Sea basin. Springer Science & Business Media]. It seems that, this is also the author of (at least) a similar study (cite and listed in the References section), using approximately the same methodology and with similar findings, for two Estonian cities (Tallinn and Tartu), one of them also analyzed in this study [Eensaar, A. (2016). Temporal and Spatial Variability of Air Temperatures in Estonia during 1756–2014. Journal of Climatology, 2016.], but an assessment of the regional climate change in surface air temperature, especially if based on long time series, may be useful for the climate change community and for people of the Baltic region.

Significance: The results were not interpreted; in fact, the manuscript does not have a “Discussion” section, which is clearly need; it is not sufficient to write that the results are interesting (line 454). The author should interpret and validate all the results (similarities and differences), with the findings of previous studies, in the region, “nearby” or in locations with similar characteristics, found for the same period, using the same or similar methodology. The author missed the opportunity to interpret/discuss several aspects such as the spatial differences of the obtained results, the “fluctuations” on annual and monthly time series. It is worth noting that the author applied statistical testes to assess the significance of the results. The conclusions are justified and supported by the results but are highly dependent on data quality.

Quality of Presentation: The article is not written in the best way. Some sections are too short (e.g., introduction, which must be enlarged with updated information to underline the importance, pertinence and novelty of the study, describe the state of the art on this subject and study area, etc.), others are extensively long (Sections 2 and 3), with unnecessary text and repetitive sentences (I will underline some examples in specific comments) and unnecessary figures, while the discussion section is missing. Another important aspect is the low quality of the figures. The author should produce figures with better quality (e.g., font type and size of title, axis, etc.). This means that the data and analyses are not presented appropriately, following the highest standards for presentation of the results.

Scientific Soundness: The author need to demonstrate that the study was correctly designed, is technically sound and the analyses performed with the highest technical standards. What does this mean? This means that the author need to demonstrate that the data and methods used in the study are appropriate and, consequently the conclusions are not affect by eventual errors. One of my major concerns is the quality of the data. Usually, long time series are not homogeneous and this characteristic is of fundamental importance in climate changes studies. The author writes several sentences (e.g., lines 55-56) as well as data preprocessing procedures (e.g., harmonization, reconstruction) which reveal that the time series could be affected by heterogeneities and, consequently, need to discuss this aspect. The author even writes that “Higher and faster rise of the air temperature in Saint Petersburg can partly be explained by the impacts of the fast growth of the city”; This means that the trend is not (only) due to climate change. It is indispensable to present the results of an analysis performed, by the author or by the data providers, to assess the homogeneity of the dataset. The dataset is small (only five time series) and there are different software packages that the author can use (e.g., HOMER, MASH), not only to detect but also to correct eventual heterogeneities in monthly time series. I do not know if this analysis was performed or not; the author must clarify this extremely important aspect. Without, this preliminary study the manuscript cannot be accepted because we need to be sure about the data quality to draw the conclusions. In my opinion the author, describes some methods with too much details. It is not necessary to explain what is a linear regression or the Student t-test; everyone knows that; however, on the other hand, it is of fundamental importance to show that you can apply the linear regression/significance test and that you applied correctly (e.g., the model should conform to all the assumptions of linear regression); for example, it does not make sense to computed a linear regression to a time series that does not present a linear increase. Please see specific comments for more details.

Interest to the Readers: I believe that this is a classical and simple climatological study. These studies should have the possibility to be published in Climate and in other journals of the scientific area. If the conclusions are correct, they are interesting for the readership of the Journal, especially, but not just, for those from the study are.

English Level: the English language is not appropriate and some parts are not even understandable; in the list of specific comment I will detail some examples, but I strongly suggest not to resubmit the manuscript before a thorough revision (professional proofreading) by a native English speaker.

Overall Merit: I believe that the article has serious flaws, the contribution is not entirely original in all aspects, does not provide a highly significant advance towards the current knowledge and the authors did not address an important long-standing question(s) with innovative experiments.

Specific comments

Taking into consideration the rating and recommendation for the publication of the manuscript, the number of specific comments should, but will not be, too large.

Line 13: all time periods should be written with a n-dash, e.g. “1961 – 1990”

The author tend to use concepts that are not appropriate in climatic research. I will just list a few number of cases:

Line 36: Climate change cannot be forecasted;

Line 42: Randomness of what? Climate system is not random;

Line 80: Specifically, which “international classifications and methods”? Proposed by WMO? Please clarify. International does not necessarily mean “internationally accepted”;

Line 136: visual assessment is subjective, subject to errors of observation/interpretation, highly dependent on the observer, etc. etc., and should therefore be avoided;

Line 150: normal distribution is characterized by the mean and variance (or standard deviation); obviously, variance and std are measures of dispersion, but why call variance by dispersion? Did you call mean/average by location?

Line 174: what do you mean by “fluctuating”? is it the range, std? Apparently, in line 177, fluctuation=anomaly!

Line 442: annual course is not common. Intra-annual variability?

Material and Methods: the data section has to be changed/redone; It is not clear/easy to follow the data origin, what were the data pre-processing procedures, who perform those procedures; which data errors affected the time series and how were corrected; what is the final data quality; the author should detailed describe the “construction” of the time series with a table where time periods, data sources/providers etc. are listed.

Line 80: What is the question?

Line 139, 149: Please explain/demonstrate why Jarque-Bera, Smirnov-Kolmogorov and Welch’s t-test can be used and are the best tests for this study; list other recent climate change studies also used these tests, for the same purpose;

Line 180: “linear trend for the entire observed period”. Why? The behaviour seems to linear for the entire period? Did you check the residuals? What did you conclude?

Line 185: For all figures and not just Figure 2. The figures are, in general, correct but rudimentary; the author should invest some time checking some articles of reference and verify how to produce more appealing figures; the title does not seem appropriate, the captions of the axes and values of the axes must have font of different size and format; etc.; for ease of reading, you should maintain the association between each location/time series and the color used to represent the values of that location, in all charts; the legends of the figure must be self-explanatory; the reader should not have to read the article to understand what is being plotted. It is not enough to say that an “average” is being represented, it is necessary to explain how this average was calculated;

Line 222: An example of repetitive text. Is it necessary to write in the results that “air temperature is regarded as a random variable”?

Line 227: “centuries” does not seems adequate; suggest 100 years and that is not the case. Please change in the entire document;

Lines 230-234: An example of too much text; It is sufficient to state that the results are statistically significant at a certain level, according to a specific test (described and justified/validated in the methodology section).

Line 226: another example of too much and repetitive text; is it necessary to write almost every time the names of the 5 cities?

Line 239: please remove “the peculiarities of”;

Line 242: replace “for Stockholm in Table 1, for Tallinn in Table 2, for Riga in Table 3, for Helsinki in Table 4, and for Saint Petersburg in Table 5” by “in Table 1 to Table 5”

Line 244: “The average annual temperatures are close in these cities.” What do you mean by close?

Line 248: Why do you present so many statistics and do not discuss all of them? You should only present results in figures/tables that you discuss in the manuscript; if you do not discuss, you do not present; in addition, only relevant results should be presented;

Lines 252, 254: Apparently, (I did not check all values), these two Tables are equal/the same. At least one of these tables is not correct; please check carefully if all figures and tables are correct;

Line 256: Which century?

Lines 265-266: results for Saint Petersburg are not listed in Table 5; please see previous comment about tables 4 and 5

Line 271: This is not clear and, apparently, not compatible with the minimum and maximum values presented in table 5; please explain;

Lines 265-266: This is the type of information that should not be in the text but in the figure caption; please correct this everywhere;

Line 275: “practically coincide” is excessive and not adequate;

Line 286: what is, exactly the question? Results section is not to pose questions but to present results; I suggest to (re)move lines 281-290;

Line 318: Are all these figures really necessary? Did you consider to move them to an annex? Did you consider to replace them by box-whisker plots? Can you explain why you prefer line instead of box-whisker plots?

Line 324: “At the same time, there have been colder January’s in the last century than in the century before”. Everywhere? If so, mention it; if not, please correct.

Line 326: “but the changes are not linear”. What do you mean by this statement? Is it only true for February?

Lines 341-349: consider remove this entire paragraph; it is not a result;

Line 345: “non-conformities are not very big”. Please avoid using subjective concepts all over the manuscript; please see also lines 460-461;

Line 351: “The results show (Figure 8)…” a different (and better, in my opinion) way to mention figures/tables; You need to uniform this procedure all over the entire manuscript;

Line 390, 406, 422, 438: Since you just present the slope, are figures 9, 10, 11 and 12 (20 figures!) really necessary?

Line 396: the trend does not seem linear; so, does it make sense to compute the linear slope? The same for other figures (e.g., in line 412).

Lines 409-410, 425-426: another examples of text that should be moved to the figure caption;

Lines 451: Figure 13 was produced to what period? In relation to what period?

Lines 460-461: “Differences in the air temperatures of the cities of the Central Baltic coastal area are relatively small”. What do you mean by this? It is not clear and subjective;

Line 502: it is quite strange the small number of citations in a climate change study; this is clearly associated with the very small size of the “Introduction” and the absence of the “Discussion” sections.

Author Response

Comments and Suggestions for Authors

Brief summary

This is a classical climate change paper. The aim is to detect, characterize and compare the long-term trend in the average surface air temperature in five locations in the Central Baltic coastal area. The author benefits from having long time series covering approximately the same period. The main contributions include the assessment of trends in annual and monthly time series, the comparison between results obtained in the different locations. However, the manuscript has several limitations and cannot be accepted in its current state.

My overall Recommendation is “Reconsider after Major Revisions” which means that he acceptance of the manuscript would depend on the revisions. The author needs to provide a point by point response or provide a rebuttal if some of the reviewer’s comments cannot be revised. Usually, only one round of major revisions is allowed. Authors will be asked to resubmit the revised paper within ten days and the revised version will be returned to the reviewer for further comments.

General comments/suggestions to the authors,

The manuscript was rated in different aspects

Originality/Novelty: The research questions are not entirely original (in the sense that the similar questions have been raised in other studies, even by the author). The results are in line with the present general knowledge (of global warming). There are other studies on this topic in the region. Please see the BACC project (BALTEX Assessment of Climate Change for the Baltic Sea Basin) [Team, B. A. (2008). Assessment of climate change for the Baltic Sea basin. Springer Science & Business Media]. It seems that, this is also the author of (at least) a similar study (cite and listed in the References section), using approximately the same methodology and with similar findings, for two Estonian cities (Tallinn and Tartu), one of them also analyzed in this study [Eensaar, A. (2016). Temporal and Spatial Variability of Air Temperatures in Estonia during 1756–2014. Journal of Climatology, 2016.], but an assessment of the regional climate change in surface air temperature, especially if based on long time series, may be useful for the climate change community and for people of the Baltic region.

Significance: The results were not interpreted; in fact, the manuscript does not have a “Discussion” section, which is clearly need; it is not sufficient to write that the results are interesting (line 454). The author should interpret and validate all the results (similarities and differences), with the findings of previous studies, in the region, “nearby” or in locations with similar characteristics, found for the same period, using the same or similar methodology. The author missed the opportunity to interpret/discuss several aspects such as the spatial differences of the obtained results, the “fluctuations” on annual and monthly time series. It is worth noting that the author applied statistical testes to assess the significance of the results. The conclusions are justified and supported by the results but are highly dependent on data quality.

Quality of Presentation: The article is not written in the best way. Some sections are too short (e.g., introduction, which must be enlarged with updated information to underline the importance, pertinence and novelty of the study, describe the state of the art on this subject and study area, etc.), others are extensively long (Sections 2 and 3), with unnecessary text and repetitive sentences (I will underline some examples in specific comments) and unnecessary figures, while the discussion section is missing. Another important aspect is the low quality of the figures. The author should produce figures with better quality (e.g., font type and size of title, axis, etc.). This means that the data and analyses are not presented appropriately, following the highest standards for presentation of the results.

Scientific Soundness: The author need to demonstrate that the study was correctly designed, is technically sound and the analyses performed with the highest technical standards. What does this mean? This means that the author need to demonstrate that the data and methods used in the study are appropriate and, consequently the conclusions are not affect by eventual errors. One of my major concerns is the quality of the data. Usually, long time series are not homogeneous and this characteristic is of fundamental importance in climate changes studies. The author writes several sentences (e.g., lines 55-56) as well as data preprocessing procedures (e.g., harmonization, reconstruction) which reveal that the time series could be affected by heterogeneities and, consequently, need to discuss this aspect. The author even writes that “Higher and faster rise of the air temperature in Saint Petersburg can partly be explained by the impacts of the fast growth of the city”; This means that the trend is not (only) due to climate change. It is indispensable to present the results of an analysis performed, by the author or by the data providers, to assess the homogeneity of the dataset. The dataset is small (only five time series) and there are different software packages that the author can use (e.g., HOMER, MASH), not only to detect but also to correct eventual heterogeneities in monthly time series. I do not know if this analysis was performed or not; the author must clarify this extremely important aspect. Without, this preliminary study the manuscript cannot be accepted because we need to be sure about the data quality to draw the conclusions. In my opinion the author, describes some methods with too much details. It is not necessary to explain what is a linear regression or the Student t-test; everyone knows that; however, on the other hand, it is of fundamental importance to show that you can apply the linear regression/significance test and that you applied correctly (e.g., the model should conform to all the assumptions of linear regression); for example, it does not make sense to computed a linear regression to a time series that does not present a linear increase. Please see specific comments for more details.

Interest to the Readers: I believe that this is a classical and simple climatological study. These studies should have the possibility to be published in Climate and in other journals of the scientific area. If the conclusions are correct, they are interesting for the readership of the Journal, especially, but not just, for those from the study are.

English Level: the English language is not appropriate and some parts are not even understandable; in the list of specific comment I will detail some examples, but I strongly suggest not to resubmit the manuscript before a thorough revision (professional proofreading) by a native English speaker.

Overall Merit: I believe that the article has serious flaws, the contribution is not entirely original in all aspects, does not provide a highly significant advance towards the current knowledge and the authors did not address an important long-standing question(s) with innovative experiments.

Specific comments

Taking into consideration the rating and recommendation for the publication of the manuscript, the number of specific comments should, but will not be, too large.

Line 13: all time periods should be written with a n-dash, e.g. “1961 – 1990”

A: Corrected

The author tend to use concepts that are not appropriate in climatic research. I will just list a few number of cases:

Line 36: Climate change cannot be forecasted;

A: The author has not said that. It is written “To forecast climate change ...”

Line 42: Randomness of what? Climate system is not random;

A: The article discusses temperature changes. But the temperature is a macro-manifestation of random processes. According to statistical physics, the temperature is proportional to the average kinetic energy of the molecules.

Line 80: Specifically, which “international classifications and methods”? Proposed by WMO? Please clarify. International does not necessarily mean “internationally accepted”;

A: Estonia is a member of the WMO, so all WMO recommendations and rules are applied.

Line 136: visual assessment is subjective, subject to errors of observation/interpretation, highly dependent on the observer, etc. etc., and should therefore be avoided;

A: Visual assessment is very necessary in the first approximation, which makes it possible to assess the nature of the changes. If it were pointless, there would be no need for drawings in scientific articles. This study is not limited to visual assessment. Using mathematical statistics methods, a very correct result is obtained.

Line 150: normal distribution is characterized by the mean and variance (or standard deviation); obviously, variance and std are measures of dispersion, but why call variance by dispersion? Did you call mean/average by location?

A: Dispersion is a measure of continuous normal distribution. However, the measured temperatures are discrete values, i.e. a sample of continuous distribution, the standard deviation calculated from them is not equal to the dispersion of continuous distribution. Calculations are made for different time periods and locations. For the clarity, it is good to get to know for example “Hennemuth, B., Bender, S., Bülow, K., Dreier, N., Keup-Thiel, E., Krüger, O., Mudersbach, C., Radermacher, C., Schoetter, R. (2013): Statistical methods for the analysis of simulated and observed climate data, applied in projects and institutions dealing with climate change impact and adaptation. CSC Report 13, Climate Service Center, Germany. 138 p.”

Line 174: what do you mean by “fluctuating”? is it the range, std? Apparently, in line 177, fluctuation=anomaly!

A: Line 174 - extent of change; Line 177 corrected.

Line 442: annual course is not common. Intra-annual variability?

Material and Methods: the data section has to be changed/redone; It is not clear/easy to follow the data origin, what were the data pre-processing procedures, who perform those procedures; which data errors affected the time series and how were corrected; what is the final data quality; the author should detailed describe the “construction” of the time series with a table where time periods, data sources/providers etc. are listed.

A:

Homogenized data means that as well as correcting inaccuracies and interpolating missing data, consideration is also given to differences that can occur due to changing instrument (measurement method) or moving the measurement site. After homogenization the entire data period should behave as if it was measured at the same location with the same instruments and methods.

Data pre-protsessing procedures are described in detail:

Moberg A, Bergström H, Ruiz Krigsman J, Svanered O. 2002: Daily air temperature and pressure series for Stockholm (1756-1998). Climatic Change 53: 171-212

Tarand A., Jaagus, J., Kallis A., Eesti kliima minevikus ja tänapäeval, Tartu Ülikooli Kirjastus, Tartu, Estonia, 2013. 631 p.

Специализированные массивы для климатических исследований (In Russian: Specialized arrays for climate research). Available online: http://aisori-m.meteo.ru/waisori/

Line 80: What is the question?

A: Line 80 - There's no question.

Line 139, 149: Please explain/demonstrate why Jarque-Bera, Smirnov-Kolmogorov and Welch’s t-test can be used and are the best tests for this study; list other recent climate change studies also used these tests, for the same purpose;

A: The prerequisite for using the Welch t-test (which is the Student's t-test variation) is that it is a normal distribution. Jarque-Bera, Smirnov-Kolmogorov test may be used to check the data distribution normality. Recommended for investigation of temperature changes in “Hennemuth, B., Bender, S., Bülow, K., Dreier, N., Keup-Thiel, E., Krüger, O., Mudersbach, C., Radermacher, C., Schoetter, R. (2013): Statistical methods for the analysis of simulated and observed climate data, applied in projects and institutions dealing with climate change impact and adaptation. CSC Report 13, Climate Service Center, Germany. 138 p.”

Buishand, T. A., and J. J. Beersma, 1996: Statistical tests for comparison of daily variability in observed and simulated climates. J. Climate,9, 2538–2550.

https://climatedataguide.ucar.edu/climate-data-tools-and-analysis/trend-analysis National Center for Atmospheric Research Staff (Eds). Last modified 05 Sep 2014. "The Climate Data Guide”.

Line 180: “linear trend for the entire observed period”. Why? The behaviour seems to linear for the entire period? Did you check the residuals? What did you conclude?

A: For natural processes, the linear trend is a very good approximation, also for annual average temperatures. For short time periods, a variety of relationships can be found. However, their reliability is much lower. The determination coefficients are given in the figures, which describes residuals.

Line 185: For all figures and not just Figure 2. The figures are, in general, correct but rudimentary; the author should invest some time checking some articles of reference and verify how to produce more appealing figures; the title does not seem appropriate, the captions of the axes and values of the axes must have font of different size and format; etc.; for ease of reading, you should maintain the association between each location/time series and the color used to represent the values of that location, in all charts; the legends of the figure must be self-explanatory; the reader should not have to read the article to understand what is being plotted. It is not enough to say that an “average” is being represented, it is necessary to explain how this average was calculated;

A: The figures are all corrected. However, neither WMO nor any other organization has determined which drawings may or may not be. In the author's opinion, they are compiled in the best possible way, which allows good understanding. The appropriate font size will depend on the size of the font they will be presented in the final. This is a simple technical problem. All figures in editable form are attached to the manuscript. 1. Average is the arithmetic mean. 2. Weighted average is similar to an ordinary arithmetic mean, except that instead of each of the data points contributing equally to the final average, some data points contribute more than others (Equation (1)). 3. Moving average  is a calculation to analyze data points by creating a series of averages of different subsets of the full data set.

Line 222: An example of repetitive text. Is it necessary to write in the results that “air temperature is regarded as a random variable”?

A: This is very important, especially because in traditional climate studies this position is considered heretic.

Line 227: “centuries” does not seems adequate; suggest 100 years and that is not the case. Please change in the entire document;

A: Corrected.

Lines 230-234: An example of too much text; It is sufficient to state that the results are statistically significant at a certain level, according to a specific test (described and justified/validated in the methodology section).

A: This is not an insignificant text. Here are the specific results of Welch's t-test on annual average temperature changes.

Line 226: another example of too much and repetitive text; is it necessary to write almost every time the names of the 5 cities?

A: Partially corrected.

Line 239: please remove “the peculiarities of”;

A: Corrected

Line 242: replace “for Stockholm in Table 1, for Tallinn in Table 2, for Riga in Table 3, for Helsinki in Table 4, and for Saint Petersburg in Table 5” by “in Table 1 to Table 5”

A: Corrected.

Line 244: “The average annual temperatures are close in these cities.” What do you mean by close?

A: Corrected – (language mistake).

Line 248: Why do you present so many statistics and do not discuss all of them? You should only present results in figures/tables that you discuss in the manuscript; if you do not discuss, you do not present; in addition, only relevant results should be presented;

A: All results are discussed in the text. If a problem remains incomprehensible to some of the researchers who do not know enough mathematical statistics, then more can be explained. But then these problems should be specifically addressed. It is a dialectical contradiction, because on the one hand it is necessary to shorten the text and write more text on the other.

Lines 252, 254: Apparently, (I did not check all values), these two Tables are equal/the same. At least one of these tables is not correct; please check carefully if all figures and tables are correct;

A: There was really a technical (copy, paste) mistake. Mistake corrected.

Line 256: Which century?

A: Corrected (20. and beginning of 21. century)

Lines 265-266: results for Saint Petersburg are not listed in Table 5; please see previous comment about tables 4 and 5

A: There was really a technical (copy, paste) mistake. Mistake corrected.

Line 271: This is not clear and, apparently, not compatible with the minimum and maximum values presented in table 5; please explain;

A: There was a technical (Table copy, paste) mistake . Mistake corrected.

Lines 265-266: This is the type of information that should not be in the text but in the figure caption; please correct this everywhere;

A: Lines 265-266 do not contain any text to handle the figures. The line numbers may be offset.

Line 275: “practically coincide” is excessive and not adequate;

A: Corrected.

Line 286: what is, exactly the question? Results section is not to pose questions but to present results; I suggest to (re)move lines 281-290;

A: Partially corrected.

Line 318: Are all these figures really necessary? Did you consider to move them to an annex? Did you consider to replace them by box-whisker plots? Can you explain why you prefer line instead of box-whisker plots?

A: Frequency allocations are important for statistical calculations. The number of figures is reduced from 12 to 4 (January, April July, October). Box-whisker plots are not suitable in this case. It would be possible to use column charts without gaps, but then there would be no five frequency distributions on one figure.

Line 324: “At the same time, there have been colder January’s in the last century than in the century before”. Everywhere? If so, mention it; if not, please correct.

A: Corrected (in Saint Petersburg only).

Line 326: “but the changes are not linear”. What do you mean by this statement? Is it only true for February?

A: The text has been deleted because there is no longer a figure for February. Here, linearity would mean that the frequency distribution has not changed significantly, but has shifted as a whole in one direction

Lines 341-349: consider remove this entire paragraph; it is not a result;

A: Title “Results” changed to title “Results and Discussion”.

Line 345: “non-conformities are not very big”. Please avoid using subjective concepts all over the manuscript; please see also lines 460-461;

A: If the value of the test statistic is lower than the predetermined significance level (in this case 0.05). As an example, the results of the St. Petersburg Jarque-Bera analysis:

Saint Petersburg

Variable\Test

Jarque-Bera

Jan

0.019646262

Feb

0.491850224

Mar

0.171393199

Apr

0.867932381

May

0.564787304

Jun

0.51808494

Jul

0.106084659

Aug

0.409860313

Sep

0.464428706

Oct

0.097568813

Nov

0.273046446

Dec

0.077212698

Average

0.529171756

January frequency distribution test-statistic is 0.01964 < 0.05. Normalization has not been proven, but also not rejected.

Line 351: “The results show (Figure 8)…” a different (and better, in my opinion) way to mention figures/tables; You need to uniform this procedure all over the entire manuscript;

A: Agree. Partially corrected.

Line 390, 406, 422, 438: Since you just present the slope, are figures 9, 10, 11 and 12 (20 figures!) really necessary?

A: Figures 9 to 12 are recreated. The Figures now show a 10-year moving average of month average air temperatures. (January, April, July and October). The former 20 figures are included in the Supplementary material.

Line 396: the trend does not seem linear; so, does it make sense to compute the linear slope? The same for other figures (e.g., in line 412).

A: The long-term trend is certainly linear. Of course, short-term trends can be found anywhere.

Lines 409-410, 425-426: another examples of text that should be moved to the figure caption;

A: Figures 9 to12 are deleted and recreated.

Lines 451: Figure 13 was produced to what period? In relation to what period?

A: Figure 13 is produced to the entire observed period.

Lines 460-461: “Differences in the air temperatures of the cities of the Central Baltic coastal area are relatively small”. What do you mean by this? It is not clear and subjective;

A: This is best seen in the frequency distributions (Figure 6) and summaries of average air temperatures Tables 2-6 (Supplementary material).

Line 502: it is quite strange the small number of citations in a climate change study; this is clearly associated with the very small size of the “Introduction” and the absence of the “Discussion” sections.

A: The introduction has been improved and the reference list has been extended.

Thank you so much for submitting your valuable review comments!

Reviewer 2 Report

December 2018,

A review on the manuscript in journal Climate entitled "Peculiarities of long-term changes in air temperatures near the ground surface in the central Baltic coastal area" written by Age Ensaa.

Recommendation: Major revision

This manuscript examined the observation surface-warming trends at five cities in the Baltic area to show the peculiarities of the trends. The author showed that the warming trend over these cities is faster than the global or Northern Hemispheric trend. Then, the author further examined the dependence of the trend on the calendar months. As a result, the warring trend is significant especially in the boreal spring season as well as the autumn, while it is not significant in the summer except for a city.

This observation-based research on the regional global-warming trend is informative and the method is described in detail very much. Although the manuscript must be improved in terms of the introduction and figures' representation, I would recommend it for the publication in the journal Climate after the major revision as described below.

Major comments: 

1, Introduction: The introduction has to describe the background of this work. However, there is no review on the seasonal dependence of the global-warming trend in detail. Actually, section 1 is too short to be the introduction. I do recommend adding the reviews on the regional climate change over Europe and Baltic region as well as the general reviews on the global warming trend seasonality (e.g.,Sparks and Menzel (2002). Int. J. Climatorol. doi:10.1002/joc.821).

2, Introduction: This manuscript is length even though the methods and results are simple relatively. After clarifying the topic and the background issue as mentioned above, adding a brief direction of this study in the introduction may help the readers to see the contents at a glance.

3, Figures: Too many panels in the manuscript. Reducing the number of the panels of figures is strongly recommended to emphasize the actual message in this work. 

Figure 2: put all the panels in a page by shrinking the panels vertically and use a single time axis consistently among the panels for ease to compare.

Figure 7: consider to make this figure look better.

Figures 9-12: should be change to be more concise since the information mentioned in the text is very small (only 13 lines!). A possible way to do so is to show only the 10-year moving lines of a month at the locations in a single panel as in Fig. 3; then there will be four panels, Jan,Apr,Jul,Oct. This can be a single figure. Then, describe the findings in the figure in a single paragraph. If the detailed data is required on purpose, it can be the supplementary figure.

4, Peculiarity of the cities: The topic indicates that the five cities in the Baltic region is peculiar. However, there is less comparison with the different regions. To emphasize the peculiarity, I would suggest showing the global or Northern Hemispheric map of the warming trend and its ratio of the spring to annual mean as a discussion. This can be done using Hadley Centre's data. This is interesting to see the locality of the trend in Baltic region.

Minor comments:

5, Methods: The method is lengthy. A part of this section can be removed or moved to the introduction (e.g., Lines 47-60). This section should focus only on the data and methods.

6, Figure 1: Add the unit for the scale.

7, Data: Adding a summary table on the observational datasets may be useful to show the data in detail.

8, Figure 3: It is interesting to show the Hadley Centre data over the Baltic region to see if the local observations are consistent with the global data in the same area.

9, Figure 5: The line for Petersburg 1901-2017 is partly dotted by mistake. Correct it.

10, Talbe 5: all the values in Table 5 is the same with Table 4. This is wrong. Correct it.

11, Lines 355-356, 361-363, 456-458, 481-483: The significant trend over Saint Petersburg in the boreal summer is interesting to readers. However, the author's interpretation is not based on any evidence nor previous studies. Show the evidence or cite proper works that can support this result. Otherwise, this part needs to be removed or clearly state that such significant trend needs a study further.

12, Figure 8: The annual mean (Year) should not be in the same lines with the calendar months. To avoid misleading presentation, use a separate plot for the annual mean, for example.

13, Figure 13: Did the author use the different data period for calculation the trend? If so, it would be recommended using the same period to compare it among the different locations.

13, Figure 13: Spell out "YA".

Author Response

Top of Form

Comments and Suggestions for Authors

December 2018,

A review on the manuscript in journal Climate entitled "Peculiarities of long-term changes in air temperatures near the ground surface in the central Baltic coastal area" written by Age Ensaa.

A: Autor name: Agu Eensaar

Recommendation: Major revision

This manuscript examined the observation surface-warming trends at five cities in the Baltic area to show the peculiarities of the trends. The author showed that the warming trend over these cities is faster than the global or Northern Hemispheric trend. Then, the author further examined the dependence of the trend on the calendar months. As a result, the warring trend is significant especially in the boreal spring season as well as the autumn, while it is not significant in the summer except for a city.

This observation-based research on the regional global-warming trend is informative and the method is described in detail very much. Although the manuscript must be improved in terms of the introduction and figures' representation, I would recommend it for the publication in the journal Climate after the major revision as described below.

Major comments: 

1, Introduction: The introduction has to describe the background of this work. However, there is no review on the seasonal dependence of the global-warming trend in detail. Actually, section 1 is too short to be the introduction. I do recommend adding the reviews on the regional climate change over Europe and Baltic region as well as the general reviews on the global warming trend seasonality (e.g.,Sparks and Menzel (2002). Int. J. Climatorol. doi:10.1002/joc.821).

A: Introduction is rewritten. The suggested article is very interesting, but because it deals with phenology (selected examples of ground flora, bird migration, tree flowering, and harvest timing), it could not be used directly. In Estonia, for example, dates of rye kelp and rye cutting have been used for historical climate analysis [11, 14].

2, Introduction: This manuscript is length even though the methods and results are simple relatively. After clarifying the topic and the background issue as mentioned above, adding a brief direction of this study in the introduction may help the readers to see the contents at a glance.

A: Introduction is rewritten. Since it is a simple scientific study, the introduction cannot be very long and deal with all aspects of climate change.

3, Figures: Too many panels in the manuscript. Reducing the number of the panels of figures is strongly recommended to emphasize the actual message in this work. 

A: The number of figures has been significantly reduced, some of them being incorporated into the supplementary material.

Figure 2: put all the panels in a page by shrinking the panels vertically and use a single time axis consistently among the panels for ease to compare.

A: Vertical figures do not make comprehension and comparison better. Figure 3 summarizes the direct comparison of changes with a single time axis..

Figure 7: consider to make this figure look better.

A_: Figure 7 is reduced to four panels : Jan, Apr, Jul, Oct. Figure 7 is reformatted.

Figures 9-12: should be change to be more concise since the information mentioned in the text is very small (only 13 lines!). A possible way to do so is to show only the 10-year moving lines of a month at the locations in a single panel as in Fig. 3; then there will be four panels, Jan,Apr,Jul,Oct. This can be a single figure. Then, describe the findings in the figure in a single paragraph. If the detailed data is required on purpose, it can be the supplementary figure.

A: Thank You! Figures 9 to 12 are recreated. The Figures now show a 10-year moving average of month average air temperatures. (January, April, July and October). The former 20 figures are included in the Supplementary material.

4, Peculiarity of the cities: The topic indicates that the five cities in the Baltic region is peculiar. However, there is less comparison with the different regions. To emphasize the peculiarity, I would suggest showing the global or Northern Hemispheric map of the warming trend and its ratio of the spring to annual mean as a discussion. This can be done using Hadley Centre's data. This is interesting to see the locality of the trend in Baltic region.

A: Global warming according to Hadley Centre’s data (both Global and Northern) is shown as a comparison for Figure 3 as well as Figure 4. From there it can be seen that warming in the Baltic coastal cities is much faster than in the Global and in Northern hemisphere.

Minor comments:

5, Methods: The method is lengthy. A part of this section can be removed or moved to the introduction (e.g., Lines 47-60). This section should focus only on the data and methods.

A: Lines 47-50 moved to Introduction. Lines 51 – 60 already deals with data and related issues (harmonization of data), this is very important when using long-term observations data.

6, Figure 1: Add the unit for the scale.

A:  The scale is added to Figure 1.

7, Data: Adding a summary table on the observational datasets may be useful to show the data in detail.

A: Summary table added (Table 1).

8, Figure 3: It is interesting to show the Hadley Centre data over the Baltic region to see if the local observations are consistent with the global data in the same area.

A: Thank You!

9, Figure 5: The line for Petersburg 1901-2017 is partly dotted by mistake. Correct it.

A: Corrected

10, Talbe 5: all the values in Table 5 is the same with Table 4. This is wrong. Correct it.

A: There was a technical (Table copy, paste) mistake. Mistake corrected. Tables moved to Supplementary material.

11, Lines 355-356, 361-363, 456-458, 481-483: The significant trend over Saint Petersburg in the boreal summer is interesting to readers. However, the author's interpretation is not based on any evidence nor previous studies. Show the evidence or cite proper works that can support this result. Otherwise, this part needs to be removed or clearly state that such significant trend needs a study further.

A:  Lines 355-356 conclusion is based on results of Welch-test analysis. Lines 361-363, 456-458 and 481-483 deleted.

12, Figure 8: The annual mean (Year) should not be in the same lines with the calendar months. To avoid misleading presentation, use a separate plot for the annual mean, for example.

A: The annual average is very important in this figure. An important result of this study is that the increase in air temperature has a seasonal shift, which is different in different months. This gives an opportunity to evaluate the time during which the warming is higher than the annual average, which is less.

13, Figure 13: Did the author use the different data period for calculation the trend? If so, it would be recommended using the same period to compare it among the different locations.

A: The trends has been calculated differently, also for same time period. The use of the entire data set gives the best estimate of the temporal change of air temperatures in the concrete city. As the calculations show, changes in trends are shrinking over time. This can be seen, for example, in annual average temperature trends (Figure 4).

13, Figure 13: Spell out "YA".

A: “YA” is changed to “A”. “YA” -Year Average. “A” – Annual avarerge.

Submission Date

07 December 2018

Date of this review

19 Dec 2018 00:06:15

Bottom of Form

© 1996-2018 MDPI (Basel, Switzerland) unless otherw

Thank you so much for submitting your valuable review comments!

Reviewer 3 Report

Comments and suggestons are given in the attachment

Author Response

Peculiarities of Long-Term Changes in Air Temperatures Near the Ground Surface in the Central Baltic Coastal Area

The author have obviously put a lot of work into the collection and organization of the data used in this study. Unfortunately, it is not easy to see what the objectives of the study are, and how the results fit the objectives, which I think, are embedded in the text.

I hope that the author will work more on this paper, and with that done, I would recommend it published. However, there is a lot of work to be undertaken. I would think that the paper should have 5 figures and 4000 words of text.

I would also change the title, peculiarities was a nice word. It remembers me of the detective novel: “The Curious Incident of the Dog in the Night-Time.”, but maybe it should address more directly the actual findings.

I have tried to make a summary, but I am not sure I am right.

1.     The author study the long-term (1850 – 2017) temperatures in five cities in the coastal Central Baltic region, Their Figure 1. There are 5 Figures, each with 5 panels that show changes in temperatures. The annual averages (Fig. 2), January temperatures, (Fig. 9), April temperatures (Fig. 10), July temperatures (Fig. 11), October temperatures (Fig12). The author divides the period 1850 to 2017 into time windows. From 1850 to 2017, the average air temperature rose between 0 and 0.1 oC /decade (was within the range?). However, for the more recent time window 1980 to 2017 the average air temperature rose with 0.5oC /decade. This is twice the average rise in northern hemisphere temperature during the same period.

2.     The air temperatures have risen most in the autumn, March and April (range 0.09oC – 0.23oC per decade). In the summer, from June to September the rise in the average temperature was lower, about 0.4 oC. The reason for the differences between temperature rises during summer and autumn is not known.

3.     One reason for the higher rise in temperature in the Baltic region versus the northern hemisphere mean may be the growth of the cities during the period 1980 to 2017 (?)

4.     Temperature change among the five Baltic cities vary (average x to x), but may be due to the proximity of the cities of St. Petersburg and Helsinki to the Arctic region and the Southern location of the other three cities (Stockholm, Tallinn and Riga).

The author remarks that the results are peculiar, but the reader have to guess why. There is no results/ information of the siting of the cities and their temperature development in the result section (But it is in the Discussion section).

A: Information on temperature changes in cities is included in Supplementary Material and Tables (which are now included in Supplementary Material). Information on temperature changes in cities is included in Supplementary Material and Tables (which are now included in Supplementary Material too). Detailed information is provided in:

Moberg A, Bergström H, Ruiz Krigsman J, Svanered O. 2002: Daily air temperature and pressure series for Stockholm (1756-1998). Climatic Change 53: 171-212

Tarand A., Jaagus, J., Kallis A., Eesti kliima minevikus ja tänapäeval, Tartu Ülikooli Kirjastus, Tartu, Estonia, 2013. 631 p.

Климат Таллина (In Russian: Tallinn Climate), Leningrad: Gidrometeoizdat, 1982, 266 p.

Климат Риги (In Russian: Riga Climate), Riga: Avots, 1983, 224 p.

In line 239, the author states the peculiarities should be ascertained. I am not sure what that means. The article would be more interesting if the author put more effort into addressing and solving and explaining some of the “peculiarities”.

A: Text corrected. The important features that a statistical analysis reveals is that:

the average temperature in the Baltic coastal cities has grown faster than the global and northern hemisphere;

the rise in temperature in different months is different, with the rise of the average temperature in the summer period has not occurred (at significance level of 0.05).

Line 142. The author state a “meaningful” hypothesis, but I do not think it is very interesting as it stands, considering the recent rise in the global temperature.

A: This is the usual procedure for checking statistical hypotheses. Specifically

Hennemuth, B., Bender, S., Bülow, K., Dreier, N., Keup-Thiel, E., Krüger, O., Mudersbach, C., Radermacher, C., Schoetter, R. (2013): Statistical methods for the analysis of simulated and observed climate data, applied in projects and institutions dealing with climate change impact and adaptation. CSC Report 13, Climate Service Center, Germany. 138 p.

There are several themes that are hinted at, and that would be potentially interesting. However, I am not that acquainted with the literature to know if it actually is interesting.

A: The Reference list has been updated by adding references that are available online.

Item # 1. What role plays the growth of the cities on temperature rise? The author states that one of the temperature stations was moved because of the growth of the city of St. Petersburg, line 362. Furthermore,youdiscussthethemeinLines454to458and483. Ifyoumadeagraphshowingthe growth of the cities as a function of time and compared that to the rise in average temperature maybe that is interesting?

A: The claim between temperature increase and the degree of urbanization has been removed. Based on the analysis of the temperature time series, the reasons for this change cannot be found.

I just googled global warming and city growth and came up with this reference: “Global surface air temperatures: Update through 1987 J Hansen, S Lebedeff - Geophysical Research Letters, 1988 - Wiley Online Library. There must be something newer.

A: The claim between temperature increase and the degree of urbanization has been removed. Based on the analysis of the temperature time series, the reasons for this change cannot be found. Even a correlative relationship may not be a causal relationship.

Item #2. What role plays the location of the cities? See lines 495 to 494.

A: The cities under investigation are located on a relatively limited area of the Baltic Sea coast. The Baltic Sea has a significant impact on the climate of these cities, as well as the opposite of the Baltic Sea.

Item #3. Does the temperature changes in the five cities correspond to the temperature changes for the northern Hemisphere? The author address this question in line 464. I think the answer is no. However, the formulation is “...have changed twice as fast...” does it mean “increased twice as much?” Is this an important finding for the assessment of northern hemisphere temperature change?

A: The air temperature of Baltic Sea coastal cities have grown faster than the northern hemisphere in the last 50 years (see Figure 3, Figure 4).

Item #4. Does the temperature records for the five cities reflect the pauses in global temperature increase? I think it would be meaningful to examine if you would find changes in temperatures within the Baltic area that correspond to the hiatus periods and the non-hiatus periods, In any way, it would be interesting at least to compare the results to the global temperature changes as you do in Figure 3, but I think it deserves a comment. Hiatus period can be found in several articles.

A: The data does not show that there is a frequent claim that “global warming stopped in 1998.” Of course it is possible to find almost any trend for a limited period via judicious choice of start and end dates of a data set that has high temporal resolution, but that is not a meaningful exercise. According to the WMO recommendations, it is not advisable to look at smaller periods of less than 30 years to find trends. In the case of hiatus, changes are usually spoken about 15 years.

Details.

Lines 31 – 40. This introduction is very general; I think it can be omitted.

A: Introduction is rewritten.

Line 58. Harmonized –reduced. It should be explained how

Homogenized data means that as well as correcting inaccuracies and interpolating missing data, consideration is also given to differences that can occur due to changing instrument (measurement method) or moving the measurement site. After homogenization the entire data period should behave as if it was measured at the same location with the same instruments and methods.

Data pre-protsessing procedures are described in detail:

Moberg A, Bergström H, Ruiz Krigsman J, Svanered O. 2002: Daily air temperature and pressure series for Stockholm (1756-1998). Climatic Change 53: 171-212

Tarand A., Jaagus, J., Kallis A., Eesti kliima minevikus ja tänapäeval, Tartu Ülikooli Kirjastus, Tartu, Estonia, 2013. 631 p.

Специализированные массивы для климатических исследований (In Russian: Specialized arrays for climate research). Available online: http://aisori-m.meteo.ru/waisori/

Line 69. Reference year

A: Corrected

Line 85. Have been reduced... did the author do this?

A: Yes! Using in [10] found functional relationship, the data of Tallinn-Maarjamäe was reduced to data Tallinn-Harku by author.

Lines 72-109. It would be nice if these results were summarized in a table. Then, it would be easier to see how and why you pick certain time windows.

A: Table added.

Line 110. You do not have to explain the t-test Line 127 Greater → great; greater than what? Line 177-184. A summarizing figure would be fine.

A: Welch  t-test is explained lines 148-164. “greater” deleted. Summarizing figure is Figure 4 (lines 220, 224).

Figures. There are too many figures, I understand that it is nice to see the temperature developments, but it is too much. They could be added to a supplementary material section. For users of the study, it would also be nice to have the actual data, so the data in numerical format would be fine in a Suppl. Mat. Section.

A: The number of figures has been significantly reduced, some of which have been included in Supplementary Material.

Line 204. The comment on the temperature rise in St. Petersburg was also made on line 181 and line 483

A: In lines 204 and 181 are the results, line 483 belongs to Conclusion. They cannot be in conflict.

Line 205 Figure 3. This is a nice figure, but the time series should be separated so that it is easier to distinguish them. You could shift the series 0.5 degrees upward, but better have different y-axes. (as they have in the journal Nature)

A: The goal is to present all the changes in one scale, which allows you to see the difference or similarity of the changes without any particular difficulty. Separately, everything is shown in Figure 2.

However, as importantly, the results obtained from the figures should be summarized in one figure or one table. This is just a suggestion. What about a graph that shows the 5 cities on the x- axis. The y- axis shows the slopes as bars. Maybe one should make 4 graphs, each showing 3 months.

A: The data are summarized in Figure 13.

Figure 5. Here one needs some help in reading the important message in the Figure.

A: Frequency distribution is the usual output of statistical analysis. See: https://en.wikipedia.org/wiki/Frequency_distribution#Construction_of_frequency_distributions

Line 245. The warmest city in the region... I think this sentence should be moved forward, to a general discussion of the temperatures in the cities. Would it be difficult to say something about the relative average temperatures based on their geographical position? You do, but rather late in the paper.

A: The claim is based on data from a tables that has been transferred to Supplementary material.

Table1, 2 etc. should be moved to supplementary material. Line 248-285. Hmm!

A: Tables moved to Supplementary material. Lines 248-285: Frequency distribution is the usual output of statistical analysis.

Figure 7. Too much and too difficult to see what it means.

A: The number of partial Figures is reduced to four. The figures show how the frequency distributions of temperatures have changed over the long term. A more accurate analysis of the changes is done with the Welch t-test.

Line 326. However, the change is not linear. I suppose this means that you have a graph for St. Petersburg with years on the x-axis and temperatures in February on the y-axis. Figure 9 shows January and Figure 10 shows April? You have a linear regression in these Figures, so it is different in February. What type of non-linearity, 2nd order polynomial?

A: There is no longer a figure for February. Here, linearity would mean that the frequency distribution (Figure 7) has not changed significantly, but has shifted as a whole in one direction. Figures 9 to 12 (the drawings themselves have been recreated, more detailed figures have been incorporated into Supplementary material) are summarized in Figure 13 (calculations are of course done for all months).. Finded relationship is not descriptive as 2nd order polynomial.

Line 329. Shifted 2oC in the positive direction +2oC is positive.

A: Correct. Shifted 2 oC in the negative direction is -2 oC.

Line 343. According → applying?

A: Corrected

Line 372. The results

A: Corrected.

Line 461. .. differ with 1.7 oC.

A: Corrected.

Line 473 and 476 Statistical analysis- you do not need this explanation

A: This is very important, especially because in traditional climate studies this position is considered heretic.

Line 488. Slightly different cities, but the nature of change, replace slightly with the actual numbers, what is the “nature of”.

A: Corrected.

Line 492. Practically absent → almost absent Table 1. To supplementary material.

A: Corrected. Tables 1 to 5 moved to Supplementary material.

Thank you so much for submitting your valuable review comments!

Round  2

Reviewer 1 Report

The author did not properly answer or followed any of the general comments/suggestions/questions, which are, by far, the most important. The author simply addressed some of the specific comments/suggestions/questions. This is not acceptable and consequently, the manuscript will not be recommended for publication.

Author Response

Thank you very much for spending your valuable time reviewing my article!

Reviewer 2 Report

The author has mostly answered my questions and, thus, the manuscript has been improved. I still feel that the figure representation could be improved and the peculiarity of the Baltic region's cities than the other regions is unclear, but this manuscript would be acceptable for the publication in Climate after minor changes related to the following comments (blue inked) to some author's replies.

>4, Peculiarity of the cities: The topic indicates that the five cities in the Baltic region is peculiar. However, there is less comparison with the different regions. To emphasize the peculiarity, I would suggest showing the global or Northern Hemispheric map of the warming trend and its ratio of the spring to annual mean as a discussion. This can be done using Hadley Centre's data. This is interesting to see the locality of the trend in Baltic region.

A: Global warming according to Hadley Centre’s data (both Global and Northern) is shown as a comparison for Figure 3 as well as Figure 4. From there it can be seen that warming in the Baltic coastal cities is much faster than in the Global and in Northern hemisphere.

From the present manuscript, readers cannot understand how the Baltic area is peculiar (as in the title of this study) than the other regions. Thus, I was wondering if such peculiarities would be similarly seen in the other cities near the Baltic region. If the other cities also have warming trends similar to the Baltic region, this study will provide a more general implication for the seasonal dependent regional warming. Thus, it seems to me that adding a large-area map to show the seasonal-dependent warming feature may increase the value of this study. If the Baltic region only has such peculiarities, it would be also very interesting, but I think other cities that have climate conditions similar to the Baltic area might also show the seasonal-dependent global warming peculiarity. The author needs to state clearly if the peculiarity is only seen in the Baltic area or may be also seen in the other cities around the world.

>8, Figure 3: It is interesting to show the Hadley Centre data over the Baltic region to see if the local observations are consistent with the global data in the same area.

A: Thank You!

I meant that adding a time series of Hadley Centre data averaged over the Baltic region to Figure 3 may emphasize the importance of the use of regional historical datasets instead of the global data to clarify the peculiarity of Baltic cities' global warming, given that the Hadley global data is too coarse to examine it. If the global data is fine enough, the present study can be easily extended to analyze all the regions around the world. If not, using the regional observational data is very critical, so it would be difficult to analyze the peculiarity globally.

>12, Figure 8: The annual mean (Year) should not be in the same lines with the calendar months. To avoid misleading presentation, use a separate plot for the annual mean, for example.

A: The annual average is very important in this figure. An important result of this study is that the increase in air temperature has a seasonal shift, which is different in different months. This gives an opportunity to evaluate the time during which the warming is higher than the annual average, which is less.

I meant that the annual average should not be connected to the monthly average lines. The annual average is not a part of the seasonal change of 12 calendar months, so connecting 13 points is misleading. Use the lines only for 12 calendar months and use separate plots or horizontal lines for the annual average.

Author Response

Reply to reviever's comments are included in the attached file.

Reviewer 3 Report

please see the comments in the PDF

Author Response

(The authors gave the same response as above.)
